# Defect-driven nanostructuring of low-nuclearity Pt-Mo ensembles for continuous gas-phase formic acid dehydrogenation

Luyao Guo[1,2,3,8], Kaixuan Zhuge[4,8], Siyang Yan[3,8], Shiyi Wang[1], Jia Zhao[4] ✉, Saisai Wang[4], Panzhe Qiao[5], Jiaxu Liu[3] ✉, Xiaoling Mou[1,6], Hejun Zhu[2] ✉, Ziang Zhao[2], Li Yan[2], Ronghe Lin[1,6] ✉ & Yunjie Ding[1,2,7] ✉

Supported metal clusters comprising of well-tailored low-nuclearity heteroatoms have great potentials in catalysis owing to the maximized exposure of active sites and metal synergy. However, atomically precise design of these architectures is still challenging for the lack of practical approaches. Here, we report a defect-driven nanostructuring strategy through combining defect engineering of nitrogen-doped carbons and sequential metal depositions to prepare a series of Pt and Mo ensembles ranging from single atoms to sub-nanoclusters. When applied in continuous gas-phase decomposition of formic acid, the low-nuclearity ensembles with unique $Pt_3Mo_1N_3$ configuration deliver high-purity hydrogen at full conversion with unexpected high activity of 0.62 $mol_{HCOOH}$ $mol_{Pt}^{-1}$ $s^{-1}$ and remarkable stability, significantly outperforming the previously reported catalysts. The remarkable performance is rationalized by a joint operando dual-beam Fourier transformed infrared spectroscopy and density functional theory modeling study, pointing to the Pt-Mo synergy in creating a new reaction path for consecutive HCOOH dissociations.

Hydrogen possesses a high energy density per mass and is widely viewed as a clean energy carrier to replace nonrenewable fossil fuels in the near future[1–4]. Albeit energy production from $H_2$ is technically effective, the storage and transportation bring tremendous challenges because of its low energy density per volume. An alternative solution to tackle this issue is to transform $H_2$ into other storage mediums that can afford on-site $H_2$ supply with a convenient chemical process. In this scenario, formic acid is identified as one of the most promising candidates owing to the unique advantages such as low cost, low toxicity, good stability, and diversity of sources, etc[5–12]. The higher volumetric energy density of formic acid than pressurized $H_2$ makes it an even more attractive fuel for vehicles. The decomposition of formic acid can occur in two different routes: dehydrogenation and dehydration. Dehydrogenation is the ideal reaction to produce $H_2$ (the byproduct

[1]Hangzhou Institute of Advanced Studies, Zhejiang Normal University, 1108 Gengwen Road, Hangzhou 311231, China. [2]Dalian National Laboratory for Clean Energy, Dalian Institute of Chemical Physics, Chinese Academy of Sciences, 457 Zhongshan Road, Dalian 116023, China. [3]Department of Catalytic Chemistry and Engineering & State Key Laboratory of Fine Chemicals, Dalian University of Technology, Dalian 116012, China. [4]Institute of Industrial Catalysis, Zhejiang University of Technology, Hangzhou 310014, China. [5]Shanghai Synchrotron Radiation Facility, Zhangjiang Lab, Shanghai Advanced Research Institute, Chinese Academy of Sciences, Shanghai 201204, PR China. [6]Key Laboratory of the Ministry of Education for Advanced Catalysis Materials, Zhejiang Normal University, 688 Yingbin Road, Jinhua 321004, China. [7]The State Key Laboratory of Catalysis, Dalian Institute of Chemical Physics, Chinese Academy of Sciences, 457 Zhongshan Road, Dalian 116023, China. [8]These authors contributed equally: Luyao Guo, Kaixuan Zhuge, Siyang Yan. ✉e-mail: jiazhao@zjut.edu.cn; liujiaxu@dlut.edu.cn; zhuhj@dicp.ac.cn; catalysis.lin@zjnu.edu.cn; dyj@dicp.ac.cn

$CO_2$ generated can be used as a useful C1 feedstock and further converted back to formic acid/formates). In the dehydration, however, undesirable CO is formed which, even at >20 ppm level, can poison the full cell catalysts for downstream applications[13]. While the dehydrogenation is energetically more favorable than the dehydration, both reactions can spontaneously proceed at room temperatures[14]. Therefore, control of the reaction kinetics is critical, which calls for rational design of highly efficient dehydrogenation catalysts.

Formic acid decomposition can proceed via photocatalysis[15,16], electrocatalysis[17,18], or thermocatalysis[19–24]. As one of the major technologies, thermocatalysis is viable both in liquid and gas phases. Liquid-phase dehydrogenation is typically performed with the presence of homogeneous (e.g., Ru, Ir, and Fe-based organometallic complexes)[25–27] or Pd-based heterogeneous catalysts[28,29]. Both systems can afford high catalytic efficiency with the turnover frequency (TOF) reaching $0.31–5.97 \, s^{-1}$ with typically negligible CO formation at low operation temperatures[13]. Nonetheless, the tedious recycling of the expensive organometallic complexes and the frequently reported deactivation of the heterogeneous catalysts due to the site blockage induced by trace CO prevent a prospective industrial process[30]. On the other hand, gas-phase dehydrogenation is attractive considering the mild exothermicity and ease of catalyst separation as compared with the endothermic liquid-phase dehydrogenation ($-15$ vs. $29 \, kJ \, mol^{-1}$)[14]. Several distinct solid catalysts have been developed for the vapor-phase decomposition of formic acid. Solymosi et al.[31] compared the performance of platinum metals deposited on different carriers (carbon, $SiO_2$, and $Al_2O_3$) and recognized Ir/C as the most active formula for CO-free $H_2$ production. Bulushev and coworkers continued the explorations of carbon-supported metal systems and identified several high-performing novel catalysts, such as K-promoted Pd/C[32], Pt/C[33], nitrogen-doped carbon (NC)-supported single-atom Pt[34], Pd[35], and Ni[20]. In addition, various carbonaceous materials-supported $\beta$-$Mo_2C$ were also explored as potential cheap replacing catalysts[22,24,36,37], which unfortunately exhibited inferior activity as compared to the platinum-group analogs. A literature survey of the developed catalysts for gas-phase formic acid decomposition reveals still insufficient activity (TOF < $0.42^{-1}$), the undesirable formation of CO particularly at full conversion, and the deteriorated performance during long-term operation[34,37–39] (Supplementary Table 1).

Nanoscale engineering of catalytic materials with precisely tailored architectures has recently emerged as a powerful strategy for sustainable technologies[40]. In particular, constructing fully exposed metal clusters[41–43] or low-nuclearity ensembles[44–46] is viewed as a promising approach to maximize the catalytic efficiency of precious metals. Further modification of these ensembles with heteroatoms can sometimes bring surprising performance due to the unexpected synergy between individual metal sites[47,48]. Yet, the fabrication of these advanced materials is still challenging. The current synthetic protocols mainly rely on (i) gas-phase/atomic layer depositions that require advanced synthesis facilities and complicated operation procedures[49], or (ii) wet chemistry methods wherein expensive metal carbonyl clusters are often used as the precursors[44], plus the difficulties in fabricating heteroatom ensembles with structural uniformity. To construct heteroatoms or multi-atom clusters, spatial confinement-pyrolysis strategy has been developed using porous materials like metal-organic frameworks or covalent-organic frameworks to prevent the sintering of different metal precursors, but is mainly limited to a few components such as Fe, Co, and Ni[48]. These drawbacks greatly restrict further exploration of this new class of inspiring materials, and thus developing alternative versatile approaches with fine-tuning of the geometric structures of low-nuclearity heteroatom ensembles is highly appealing.

In our quest for high-performing metal-supported catalysts, nitrogen-doped carbon (NC) has been adopted as a platform to engineer the atomic structures of different precious metals[50–53]. The rich surface chemistry plus tunable nitrogen defective sites allows elegant tuning of the coordination environments of the single-atom metal centers via thermal-driving surface migration of the metal species, which are then captured and stabilized by the distinct cavities on the host[51]. Following a similar philosophy, herein, we report a more advanced yet facile strategy to design a series of highly sintering-resistant low-nuclearity Pt-Mo ensembles through defect engineering of the polyaniline-derived NC hosts. The key idea lies in that, Mo atoms are first strongly coordinated by the N defects followed by the deposition of Pt, wherein controlled aggregation of Pt is realized by simply reducing the number of nitrogen defects. Contrasting to the previous methods that generally lack control of the ensemble sizes[42,54], delicate tuning with a narrow size distribution in the subnano regime (from dual atoms to ca. 0.66 nm) is realized, as unveiled by in-depth microscopic and spectroscopic characterizations. Furthermore, the newly developed pyridinic-N-coordinated $Pt_3Mo_1N_3$ ensembles delivered unprecedented activity with a reaction rate of 0.62 $mol_{HCOOH}$ $mol_{Pt}^{-1} \, s^{-1}$, high-purity $H_2$ at full conversion, and outstanding long-term stability in the continuous gas-phase formic acid decomposition. The remarkable performance, as revealed by combining operando dual-beam Fourier transformed infrared spectroscopy (DB-FTIR) and density functional theory (DFT) calculations, roots at the Pt-Mo synergy that provides an alternative route with lower energy barriers for the sequential disassociation of HCOOH. Our study provides a novel scaffold for engineering complex size-specific heteroatom metal catalysts and takes a step forward in the practical utilization of formic acid-derived hydrogen.

## Results and discussion

### Low-nuclearity Pt-Mo ensembles with defined carbon hosts

The NC hosts with tunable N defects were prepared by the pyrolysis of self-made polyaniline derived from the established oxidative polymerization method[51]. The N contents of 2.06 and 5.93 wt.% were obtained by moderating the annealing temperatures at 1073–1473 K (Supplementary Table 2). Another NC host with a higher N content of 9.4 wt.% was derived from our recently developed "ring opening-pyrolysis" approach[55]. These hosts were hereafter coded as $NC_x$ ($x$ = 0.02, 0.07, and 0.13), wherein $x$ denoted the molar N:C ratios that were determined by both elemental analysis and X-ray photoelectron spectroscopy (XPS, Supplementary Table 2). $N_2$ sorption isotherms of $NC_x$ presented the H4 shapes typical for the micropore-rich carbon-based materials with the specific surface areas ($S_{BET}$) of 464–571 $m^2 \, g^{-1}$ (Supplementary Fig. 1). In addition, transmission electron microscopic (TEM) analysis revealed the worm-like morphologies of $NC_x$, and the presence of more graphitic structures at the edges on $NC_{0.02}$ (Supplementary Fig. 2), hinting the higher graphitization probably due to the elevated pyrolysis temperature applied. The as-prepared $NC_x$ was adopted to accommodate the mono- and bimetallic Mo and/or Pt species following the strategies presented in Fig. 1. The mono-metallic catalysts were prepared by incipient wetness impregnation and subsequent calcination and reduction, while the bimetallic analogs, sequential impregnation and calcination of the Mo and Pt precursors were applied before the final reduction. Details on the synthesis and all characterization techniques employed are provided in the Methods. Inductively coupled plasma-optical emission spectroscopic (ICP-OES) analyses confirmed close contents to the nominal set values for both metals (Supplementary Table 2), and similar hysteresis loops to those of $NC_x$ were verified after deposition of the metals (Supplementary Fig. 3). The powder X-ray diffraction (PXRD) patterns exhibited two broad peaks related to the amorphous carbon hosts and only a fainted peak at $2\theta = 39.4^{\circ}$ corresponding to the Pt(111) facet was detected for Pt-Mo/$NC_{0.02}$ (Supplementary Fig. 4), suggesting the high dispersion of these metals. Agreeing with these results, no clear nanoparticles could be observed by TEM on all the catalysts, except for Pt-Mo/$NC_{0.02}$ (Supplementary Fig. 5). High-angle annular dark-field scanning

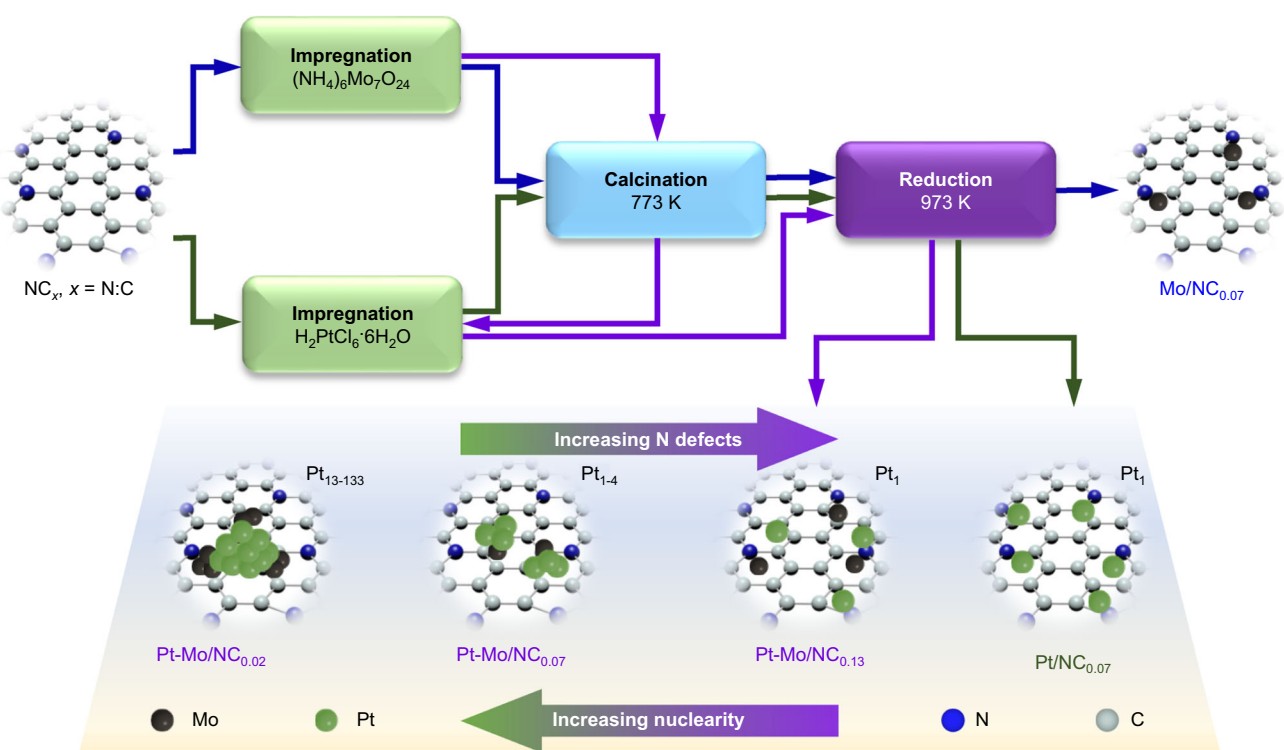

**Fig. 1 | Schematic illustrations for the catalyst synthesis.** The NC-supported mono- and bimetallic catalysts were prepared by the impregnation methods, wherein sequential impregnation was adopted for the latter. The metal ensemble sizes were tuned by varying the numbers of nitrogen defects of the carriers.

transmission electron microscopy (HAADF-STEM) was adopted to visualize the metal dispersion (Figs. 2a–e and Supplementary Fig. 6). Many white dots decorating on the gray backgrounds were observed for the mono-metallic catalysts. Statistic counting above 500 ensembles suggested the average sizes of $0.13 \pm 0.11$ and $0.22 \pm 0.14$ nm, respectively, for Pt/NC$_{0.07}$ and Mo/NC$_{0.07}$, hinting predominantly atomic-level dispersion of the metals. The role of the nitrogen defects in tuning the ensemble sizes was exemplified in the bimetallic series. Pt-Mo/NC$_{0.07}$ showed a broadened size distribution centered at 0.25 nm with the formation of more clusters, as compared with the mono-metallic counterparts. These ensemble sizes were further tunable within the subnano regime (between $0.17 \pm 0.09$ and $0.66 \pm 0.76$ nm, for Pt-Mo/NC$_{0.13}$ and Pt-Mo/NC$_{0.02}$, respectively) by simply varying the numbers of the N defects. The high density of N defects was likely responsible for maintaining the atomic dispersion for Pt-Mo/NC$_{0.13}$. On the contrary, elemental color mapping of Pt-Mo/NC$_{0.02}$ with reduced N defects revealed more homogeneous distribution of C, N, and Mo than Pt, unambiguously pointing to the clustering of the Pt atoms (Supplementary Fig. 7). By adopting a spherical model with Pt radius of 0.139 nm, the average numbers of the atoms within the clusters were estimated to be 1-4 and 13-133, respectively, for Pt-Mo/NC$_{0.07}$ and Pt-Mo/NC$_{0.02}$. Considering the high-temperature reduction treatment (973 K), our approach provides a straightforward strategy to design highly sintering-resistant low-nuclearity bimetallic ensembles with tunable and narrow size distributions.

X-ray absorption spectroscopy (XAS) in junction with XPS was further applied to unveil the electronic and geometric properties of the developed key materials. Comparison of the normalized X-ray absorption near edge structures (XANES) revealed quite different features between the mono-metallic catalysts and the respective metal foils (Fig. 2f, g). Pt/NC$_{0.07}$ displayed a pre-edge at 11570.3 eV with an upshift of 2.1 eV as compared to Pt foil and an increased whiteline intensity (1.61 vs. 1.36, Fig. 2f). Mo/NC$_{0.07}$ also exhibited a pre-edge at 20006.3 eV that was absent for Mo foil (Fig. 2g). These results confirmed the ionic feature of the metal species, i.e., Pt$^{2+}$ and Mo$^{6+}$, which

were further corroborated by the XPS showing perfect doublets centered at 72.5 and 232.6 eV, respectively, for Pt $4f_{7/2}$ and Mo $3d_{5/2}$ (Supplementary Fig. 8). The absence of surface Pt$^0$ in Pt-Mo/NC$_{0.02}$ with evidenced Pt clusters might be explained by the quantum size effect which is most prominent for small nanoparticles[56,57]. Fittings of the Fourier transformed (FT) extended X-ray absorption fine structures (EXAFS) of the mono-metallic catalysts indicated the predominant Pt−N scattering path for Pt/NC$_{0.07}$ and both Mo−O and Mo−N contributions for Mo/NC$_{0.07}$ (Fig. 2h, i and Supplementary Table 3). In comparison to the mono-metallic catalysts, the pre-edge of Pt-Mo/NC$_{0.07}$ in Pt $4f$ XANES spectrum was downshifted by 0.8 eV with a slightly reduced whiteline intensity while the Mo $3d$ XANES spectrum remained essentially identical. Besides, the Pt $4f$ and Mo $3d$ XPS spectra also only slightly downshifted by 0.1 eV. These observations jointly suggested the ionic nature of both metals in the bimetallic catalyst, probably increased electron density for the Pt species. Indeed, the Pt $4f$ FT EXAFS spectrum of Pt-Mo/NC$_{0.07}$ differed greatly from that of Pt/NC$_{0.07}$ with the evidenced scattering paths of Pt−Mo and Pt−Pt bonds, which therefore explained the increased electron density of the Pt species and their agglomeration phenomenon.

Nitrogen defects played an important role in anchoring the metal sites, which was evidenced by the presence of Pt−N bonds detected in mono- and bimetallic catalysts as well as the increasing sizes of the bimetallic Pt−Mo ensembles at decreasing N dopant. To further discriminate the contribution of different N functionality, the N $1s$ XPS spectra of Pt-Mo/NC$_x$ was assessed (Supplementary Fig. 9). Spectra deconvolution revealed the increasing shares of planar N functionalities (both pyridinic- and pyrrolic-N) at higher doping levels. This well corroborated with the inversed trend in the ensemble sizes of the supported catalysts, thus suggesting the critical roles of planar N defects in stabilizing the metal species. To understand how the N defects can coordinate with the Mo atoms, the formation energies ($E_f$) as well as Bader charges of Mo species stabilized by multiple pyridinic-N defects were calculated by DFT (Supplementary Fig. 10). A higher $E_f$ was observed with the increasing numbers of the N anchor in the MoN$_x$

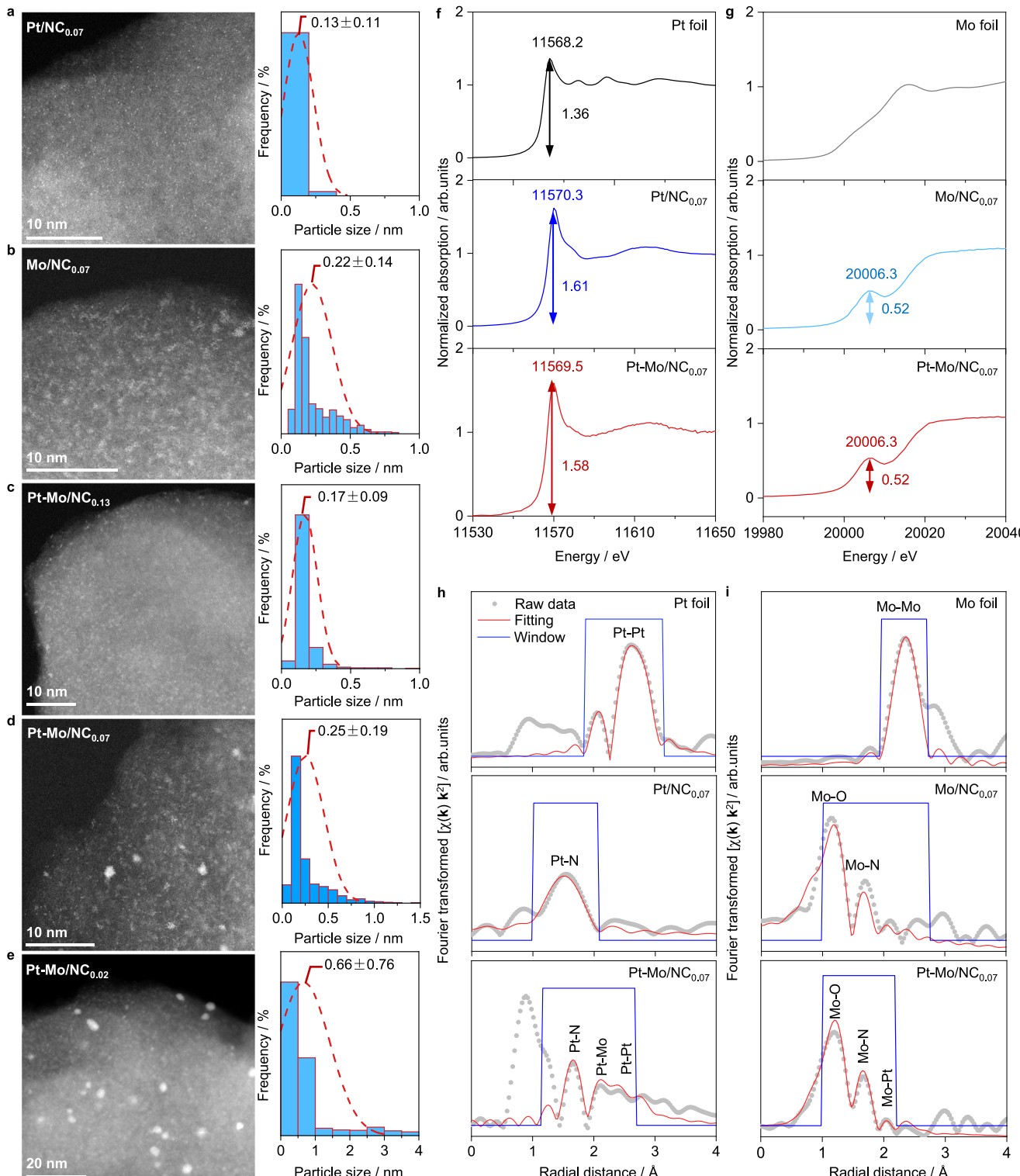

**Fig. 2 | Characterization of the key catalysts. a–e** HAADF-STEM images with the particle size distributions. The dashed lines show the fitted results, accompanied with the averaged particle sizes. **f, g** Normalized XANES spectra with the Pt and Mo foils references. The arrows show the Whiteline intensities of the Pt and Mo edges of different samples. **h, i** EXAFS spectra, and the fitting results.

entities, suggesting the higher stability. Bader charge analysis revealed the electron-deficient states of the Mo atoms in all these configurations, agreeing well with the Mo 3$d$ XPS and XANES observations for the Mo/NC$_{0.07}$ and Pt-Mo/NC$_{0.07}$. To further address the impact of the numbers of N defect on the coordination chemistry of the Mo species, two additional Mo/NC$_x$ catalysts (5 wt.% Mo, $x = 0.004$ and 0.02) were prepared as references following the same recipe for Mo/NC$_{0.07}$.

HAADF-STEM images clearly showed that a few particles were formed on Mo/NC$_{0.02}$ and more severe aggregation occurred for Mo/NC$_{0.004}$ (Supplementary Fig. 11). The lattice fringes corresponding to the (101) facets of Mo$_2$C were verified for Mo/NC$_{0.004}$. These observations were further corroborated by PXRD and Mo 3$d$ XPS analyses (Supplementary Fig. 12). Specifically, several diffraction facets corresponding to Mo$_2$C were observed on Mo/NC$_{0.004}$, and the contribution of surface

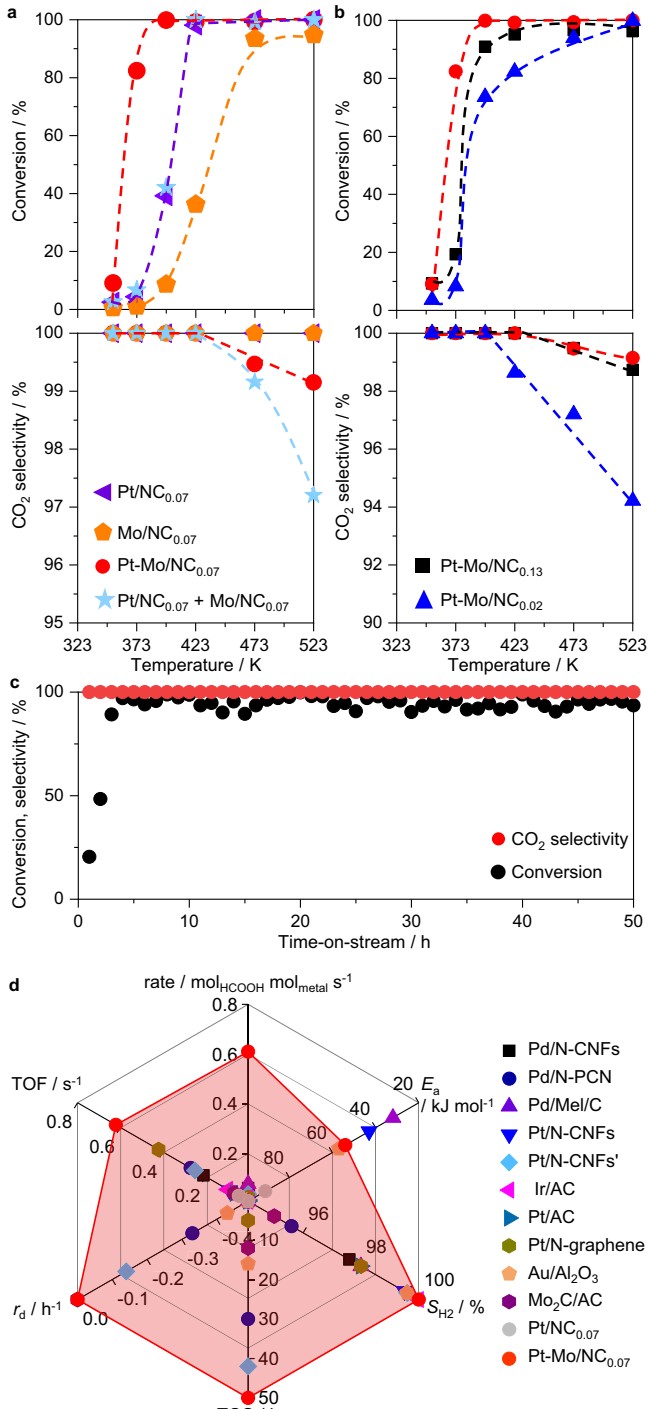

**Fig. 3 | Performance of different catalysts in formic acid decomposition.**
**a**, **b** The conversion and $CO_2$ selectivity as a function of bed temperature. Dashed lines were used to guide the eye. **c** Long-term stability performance of Pt-Mo/$NC_{0.07}$ at 373 K. **d** Comparison of the performance descriptors of Pt-Mo/$NC_{0.07}$ with those of the previously reported catalysts. The pink shaded area connects the optimal values for Pt-Mo/$NC_{0.07}$. $E_a$: apparent activation energy, TOF: turnover frequency, $r_d$: deactivation rate, TOS: time-on-stream, $S_{H2}$: selectivity of $H_2$. Details for the reference catalysts are provided in Supplementary Table 1.

$Mo^{2+}/Mo^{4+}$ species was also confirmed for this sample. Altogether, the above findings confirmed the critical role of planar N defect in stabilizing the Mo species against sintering.

Finally, the possible configuration of the metal ensembles in Pt-Mo/$NC_{0.07}$ was then explored by DFT simulations. Based on the above

thorough characterizations, ten different optimized stable models were derived (Supplementary Fig. 13) considering i) the maximal metal numbers of four ($Pt_2Mo_1$ or $Pt_3Mo_1$), and the presence of ii) Pt-Mo bonds and iii) Pt-N bonds related to $sp^2$-hydridized N defects (both pyridinic- and pyrrolic-N). Comparison of the formation energy of the metal clusters and their binding energy on various defects revealed that the pyridinic-N-stabilized $Pt_3Mo_1N_3$ with a stable tetrahedron configuration of the metal atoms possessed the highest formation energy of −16.84 eV and a high binding energy of −7.14 eV. Therefore, $Pt_3Mo_1N_3$ ensemble was later adopted for further mechanistic investigations (vide infra). In addition, we have tentatively compared the $E_f$ of $Pt_4N_3$ and $Pt_3Mo_1N_3$ with similar tetrahedron configurations (Supplementary Fig. 14). The results showed that $Pt_3Mo_1N_3$ possesses a slightly higher $E_f$ than $Pt_4N_3$ (−16.84 vs. −16.34 eV), suggesting that the presence of Mo is thermodynamically beneficial for the formation of low-nuclearity Pt-Mo ensembles.

**Catalytic performance in formic acid decomposition**
The performance of the developed low-nuclearity catalysts in the continuous gas-phase decomposition of formic acid was then evaluated in a home-made fixed-bed quartz reactor at ambient pressure (Supplementary Fig. 15). The catalytic data were first collected in a temperature-ramping mode from 353–523 K at a total gas hourly space velocity (GHSV) of 15,000 $cm^3$ $g_{cat.}^{-1}$ $h^{-1}$. At first, the performance of the $NC_{0.07}$-supported catalysts was compared, which clearly demonstrated the superior activity of Pt-Mo/$NC_{0.07}$ over Pt/$NC_{0.07}$ and Mo/$NC_{0.07}$ (Fig. 3a). Namely, full conversion was reached at a low-temperature of 398 K over Pt-Mo/$NC_{0.07}$, whereas it was delayed to 423 K on Pt/$NC_{0.07}$ and above 523 K on Mo/$NC_{0.07}$. Meanwhile, the $CO_2$ selectivity remained nearly 100% over all three catalysts at <423 K, with trace CO (selectivity <1%) formed on Pt-Mo/$NC_{0.07}$ but only at elevated temperatures. These promising results strongly hinted at the synergistic effect between the Pt and Mo species in the bimetallic catalyst. To verify this point, another contrast experiment was performed, wherein two physically mixed mono-metallic catalysts (Pt/$NC_{0.07}$ + Mo/$NC_{0.07}$) comprising the same metal loadings as those of the bimetallic analog were packed into the reactor and evaluated. This mixture exhibited similar activity to that of Pt/$NC_{0.07}$ but with poorer $CO_2$ selectivity at >423 K. The contrast test thus highlighted the importance of intimate contact and/or certain geometric structures between the Pt and Mo species in boosting the reaction rate. Next, the impact of the bimetallic ensemble sizes on the catalytic performance was examined by using the Pt-Mo/$NC_x$ series (Fig. 3b). All these catalysts exhibited similar activity profiles with steep curves observed between 373–423 K, but the temperatures corresponding to full conversion were delayed for the other two catalysts as compared with that of Pt-Mo/$NC_{0.07}$. Furthermore, the $CO_2$ selectivity as a function of the temperature on Pt-Mo/$NC_{0.13}$ was close to that on Pt-Mo/$NC_{0.07}$, while CO was formed already at 398 K and increased more rapidly at higher temperatures on Pt-Mo/$NC_{0.02}$. Our results suggested a critical size in the subnano regime for the maximized activity, which was likely linked to the unique geometry of Pt-Mo/$NC_{0.07}$ as simulated by DFT.

Given the remarkable low-temperature activity of Pt-Mo/$NC_{0.07}$, the long-term stability performance at 373 K was further evaluated in a 50 h time-on-stream (Fig. 3c). The activity gradually increased after a few hours of stabilization and slightly fluctuated at *ca.* 90–96%, while trace CO of 15.5 ppm in average was detected in our gas chromatography (Supplementary Fig. 16). Pt-Mo/$NC_{0.07}$ after the stability test was thoroughly characterized by different techniques. PXRD and Raman spectra analyses revealed the same amorphous nature of the spent catalyst and no significant alternation of the carbon carrier (Supplementary Fig. 17). No diffractions of Pt- and/or Mo-related compounds were detected by PXRD, suggesting that these metal species remained highly dispersed. This was further corroborated by HAADF-STEM observations, showing almost homogeneous

distribution of small ensembles (the white spots) scatting around the hosts (Supplementary Fig. 18). Statistic counting revealed very close metal ensemble sizes as compared to those of the fresh catalyst (0.23 ± 0.21 vs. 0.22 ± 0.14 nm), thus indicating the high stability of the bimetallic ensembles. A further survey of the core-level XPS spectra of Pt $4f$, Mo $3d$, and N $1s$ also revealed negligible differences after catalysis (Supplementary Fig. 19). All these characterizations thus demonstrated the superior robustness of the bimetallic Pt-Mo catalyst.

To better evaluate the potential of Pt-Mo/NC$_{0.07}$, a detailed comparison of the key performance descriptors with those of previously reported catalysts for gas-phase formic acid decomposition was made taking into account the stability and kinetic studies (Fig. 3d, Supplementary Fig. 20, and Supplementary Table 1). Pt-Mo/NC$_{0.07}$ possessed a moderate apparent activation energy ($E_a$) of 54 kJ mol$^{-1}$, which was much lower than those of Pt/NC$_{0.07}$ and Pt-Mo/NC$_{0.02}$ (65 and 91 kJ mol$^{-1}$, respectively). Notably, the reaction rates of Pt-Mo/NC$_{0.07}$ at 373 and 388 K reached 0.31 and 0.62 mol$_{HCOOH}$ mol$_{Pt}^{-1}$ s$^{-1}$, respectively, significantly outperforming the-state-of-the-art precious metal-based catalysts by ca. one order of magnitude under similar reaction conditions comprising the dilute gas-phase reactions. This remarkable utilization efficiency of precious metals reflected the promising industrial prospect. Furthermore, owing to the maximized exposure of the Pt atoms, Pt-Mo/NC$_{0.07}$ also stood out as the most active catalysts even by comparing the TOF based on the active surface metals. In addition, the full H$_2$ selectivity and no activity deterioration in the long-term evaluation further highlighted the outstanding performance.

### Kinetic insights from operando DB-FTIR

To shed light on the high performance of Pt-Mo/NC$_{0.07}$, the adsorption of formic acid molecules on the key catalysts was studied by employing DB-FTIR. The advantages of DB-FTIR over conventional single-beam FTIR could be explained by the fact that the former can simultaneously collect the catalyst sample and the reference spectra. As such, it could often provide more precise structural fingerprints and high-quality spectra, as have been demonstrated in the previous works[58–61]. To our delight, this superior technique was able to tackle the adsorption of formic acid on our carbon-based materials that were otherwise extremely challenging for the conventional IR. The adsorption of pure formic acid on the selected catalysts at room temperature was first studied (Supplementary Fig. 21). Pt/NC$_{0.07}$ with predominant Pt single atoms exhibited two weak adsorption bands centered at ca. 1718 and 1595 cm$^{-1}$, which were assigned to the C−O vibration of the molecularly adsorbed HCOOH (HCOOH$_{ad}$) and the O−C−O vibration due to the adsorbed formate species (HCOO$_{ad}$)[15,31]. These bands were also detected on Mo/NC$_{0.07}$ and Pt-Mo/NC$_{0.07}$ but with much stronger intensities, suggesting favorable adsorption of HCOOH on the latter. By comparing the intensities of these typical bands ($I_{1595}$:$I_{1718}$), one can find a much higher ratio for Pt-Mo/NC$_{0.07}$ over Mo/NC$_{0.07}$. This might suggest the higher dissociation propensity of formic acid molecules on Pt-Mo/NC$_{0.07}$. In contrast, Pt-Mo/NC$_{0.02}$ did not exhibit obvious spectroscopic features at 1800−1400 cm$^{-1}$.

To acquire in-depth kinetic insights, operando DB-FTIR was performed to study the adsorption and activation of HCOOH on Pt-Mo/NC$_{0.07}$ as well as the respective mono-metallic catalysts (Fig. 4). Besides the two most intense bands at ca. 1718 and 1595 cm$^{-1}$, another two bands at 1340 and 1190 cm$^{-1}$ corresponding to the symmetric O−C−O vibration in the HCOO$_{ad}$ species and the C−O vibration in molecular HCOOH[15,31] were detected at room temperature on all the three catalysts (Fig. 4a). These bands gradually attenuated to different degrees at increasing temperatures. As exemplified in the zoomed spectra (Fig. 4b), the bands at ca. 1718 cm$^{-1}$ gradually disappeared on all the catalysts, whereas the band at 1595 cm$^{-1}$ showed divergent evolution on the different catalysts. These bands greatly attenuated at high temperatures for Mo/NC$_{0.07}$ and Pt-Mo/NC$_{0.07}$ but remained

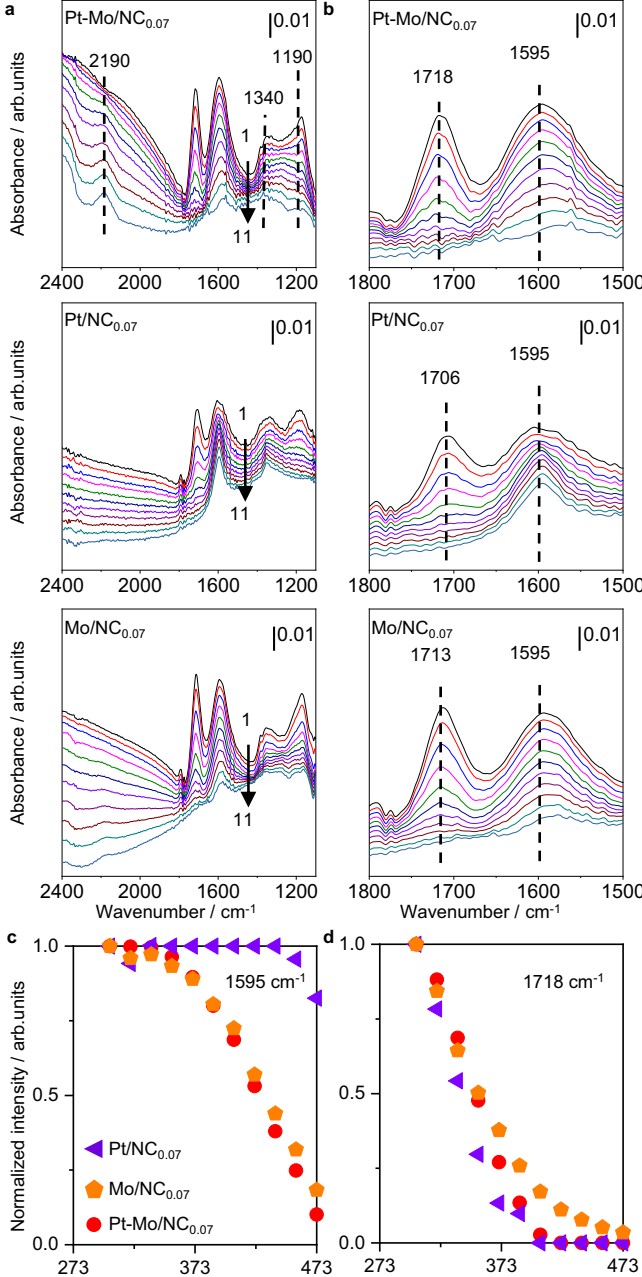

**Fig. 4 | Kinetic fingerprints of the key catalysts. a, b** Operando DB-FTIR spectroscopy with formic acid adsorbed on the mono- and bimetallic catalysts at elevated temperatures from room temperature to 473 K (1 → 11, as indicated by the arrows). The typical absorption bands were indicated by the dashed lines. Normalized intensities of **c** 1595 cm$^{-1}$, and **d** 1718 cm$^{-1}$ bands as a function of the temperature; all the spectra were referred to those collected at room temperature.

essentially stable on Pt/NC$_{0.07}$. To better compare these features, the normalized band intensities over different catalysts were presented (Fig. 4c, d). The band related to HCOOH$_{ad}$ at 1718 cm$^{-1}$ disappeared at a comparably low temperature of 405 K on Pt/NC$_{0.07}$ and Pt-Mo/NC$_{0.07}$ and at about 473 K on Mo/NC$_{0.07}$. On the contrary, the desorption of HCOO$_{ad}$ at 1595 cm$^{-1}$ remarkably decreased with temperature on Pt-Mo/NC$_{0.07}$ and Mo/NC$_{0.07}$, while on Pt/NC$_{0.07}$ it only mildly decreased by about one quarter at 473 K. Furthermore, an additional band at 2190 cm$^{-1}$, probably associated with CO vibration at electron-deficient Mo species[62], gradually built up at higher temperatures on the bimetallic catalysts but was absent on the other two mono-metallic

catalysts. This hinted at the accelerated dehydration of HCOOH on the bimetallic catalyst, thus agreeing well with their catalytic testing results. In general, the above DB-FTIR study evidenced the different kinetic fingerprints of formic acid molecules with distinct mono- and bimetallic sites. On the representative single-atom Pt sites, strong adsorption of formic acid was observed, but the deprotonation of the second H atom in $HCOO_{ad}$ was much more difficult. The mono-metallic Mo catalyst also showed a higher propensity toward HCOOH adsorption and its dissociation but displayed the poorest decomposition activity. This suggested that other fundamental steps, such as the abstraction of another H atom, might be more energy-demanding than the dissociation of HCOOH (vide infra). Another explanation could be the too-strong adsorption of HCOOH on the $Mo^{6+}$ sites as indicated by the much higher desorption temperatures in the operando DB-FTIR study. In contrast, these two fundamental steps of formic acid adsorption and dissociation were both more favorable on the low-nuclearity Pt-Mo ensembles, thus accounting for the highest activity.

## Mechanistic insights from DFT calculations

Platinum has been recognized as one of the best components among the platinum-group metals for the gas-phase decomposition of formic acid. The seminal works by Bulushev et al. pointed at the geometric and electronic effects of N-doping of the carbon hosts on the Pt nanoclusters (1–2.3 nm)[33,34], leading to significantly improved decomposition activity. In particular, they found that the single Pt atoms anchored by a pair of pyridinic-N exhibited superior catalytic performance than the analogs supported on the N-free hosts. In our study, the impact of molybdenum modification on the catalytic performance of different Pt ensembles in the subnano region, induced by defect-driving nanostructuring, was disclosed. The promotional effect was more prominent on the low-nuclearity Pt than the single atoms and bigger nanoclusters. In addition, our contrast experiment demonstrated the importance of intimate contact between Pt and Mo for efficiently activating formic acid molecules. These results thus highlighted the size effect and the synergistic catalysis of the Pt-Mo ensembles.

Since our operando DB-FTIR study evidenced the different interactions of formic acid on the distinct mono- and bimetallic sites, to gain further insights into the Pt-Mo synergy in Pt-Mo/NC$_{0.07}$, the reaction mechanisms of HCOOH decomposition on different model catalysts were studied by DFT simulations. To examine the nuclearity effects of the Pt catalysts, different model systems were considered: $Pt_1N_4$ (single Pt atom stabilized by four pyridinic-N in a planar mode), $Pt_3Mo_1N_3$, and Pt(111). We first compared the adsorption and dissociation energies of HCOOH on these model systems, both displaying the same trends with the order of $Pt_3Mo_1N_3 > Pt(111) > Pt_1N_4$ (Supplementary Fig. 22). This suggested that the activation of HCOOH on $Pt_3Mo_1N_3$ was thermodynamically favorable over the other two systems. Furthermore, the positive dissociation energy on $Pt_1N_4$ hinted at the difficulty in deprotonation of HCOOH molecules. These results were therefore fully consistent with the DB-FTIR observations. The reaction paths of HCOOH decomposition on these catalyst models were further simulated (Fig. 5 and Supplementary Fig. 23). The reaction coordinate on Pt(111) showed a small energy barrier of 0.30 eV for the adsorption of HCOOH, which underwent an exothermic H dissociation step to generate surface *HCOO species adsorbed in a bridge configuration. The dissociation of the second H in *HCOO for $CO_2$ release was most energy-demanding with a barrier of 1.55 eV. Lastly, the combination of two adsorbed H species to release molecular $H_2$ needed to overcome another barrier of 1.12 eV. On the contrary, the deprotonation of HCOOH at $Pt_1N_4$ should overcome a very high energy barrier of 2.60 eV, suggesting that the activation of HCOOH at $Pt_1N_4$ was also kinetically unfavorable. To probe the Pt-Mo synergy at $Pt_3Mo_1N_3$, two different reaction paths were considered. When HCOOH was first adsorbed at the Pt site (Supplementary Fig. 24), the H atom in the

hydroxyl groups was subtracted and bonded on the Mo sites, accompanied by the adsorption of *HCOO through two O atoms at the top Pt sites. These steps were strongly exothermic (−2.73 eV) without any barriers. The dissociation of the second H atom in *HCOO was most energy-demanding (2.70 eV), which needed to fist break one Pt−O bond, and then transfer the second H to the Mo sites with the release of $CO_2$. In addition, the combination of *H should surpass another barrier of 1.17 eV. The significantly higher energy barrier than that of Pt(111) was contradictory to the catalytic activity of the low-nuclearity Pt-Mo/NC$_{0.07}$ and thus suggested the possibility of other paths. Therefore, HCOOH firstly adsorbed at the Mo sites was considered (Fig. 5). The dissociation of the H atom in the hydroxyl groups was spontaneous and thermodynamically more favorable than on Pt(111) (reaction heat 0.76 vs. 0.40 eV). In this case, the H atom was bonded with two Pt atoms while *HCOO was adsorbed at Mo and the top Pt sites through bonding with two O atoms. This unique configuration was quite beneficial for the activation of the second H in *HCOO, wherein the H atom was transferred to the top Pt sites with a lower barrier as compared with that on Pt(111) (1.18 vs. 1.55 eV). To further support the synergistic effect, DFT simulations on the other mono-metallic sites, i.e., $MoN_3$ (single Mo atom coordinated with triple pyridinic-N sites) and $Pt_4N_3$ (tetrahedron Pt clusters stabilized in triple pyridinic-N sites) were performed. Comparison of the reaction coordinates revealed higher energy barriers for (i) the activation of *HCOO and desorption of *COO on $MoN_3$ (Supplementary Fig. 25), and (ii) the deprotonation of HCOOH on $Pt_4N_3$ (Supplementary Fig. 26), as compared with the respective elemental steps on the $Pt_3Mo_1N_3$ sites. To further substantiate the theoretical findings, a Mo-free reference – Pt/NC$_{0.02}$ with similar particle size distributions (0.31 ± 0.27 nm, Supplementary Fig. 27) as those of Pt-Mo/NC$_{0.07}$ was prepared and evaluated in HCOOH decomposition, which indeed displayed lower activity and much poorer $CO_2$ selectivity (Supplementary Fig. 28). Based on the previous works on Pt single crystals[63] and detailed kinetic and modeling study on Pd/C catalysts[64], the dissociation of formic acid molecules into formate species was proposed to be the rate-determining step. Our DB-FTIR experiments and DFT modeling studies demonstrated the cooperative catalysis between Pt and Mo in $Pt_3Mo_1N_3$ for lowering the overall energy barriers in HCOOH decomposition as compared with $Pt_1N_4$ and Pt(111), which might explain the superior catalytic performance.

In summary, we have successfully designed a new class of bimetallic platinum and molybdenum ensembles supported on nitrogen-doped carbon (NC) via a straightforward impregnation-reduction approach. Our defect-driven nanostructuring strategy coupling manipulation of the N defects and sequential metal depositions can systematically alter the geometric distribution of Pt species from single atoms to sub-nanoclusters. A high number of planar N defects was favorable for stabilizing the atomic dispersion of the Pt species, while accelerated agglomeration occurred at reduced defect numbers. The developed bimetallic catalysts together with the respective mono-metallic analogs were evaluated for the continuous gas-phase decomposition of formic acid to generate high-purity hydrogen. The low-nuclearity Pt-Mo ensembles (Pt-Mo/NC$_{0.07}$) displayed unprecedented high activity with a reaction rate of 0.62 mol$_{HCOOH}$ mol$_{Pt}^{-1}$ s$^{-1}$, significantly outperformed (i) the bimetallic analogs featuring single Pt atoms or bigger Pt sub-nanoclusters, (ii) the respective mono-metallic catalysts, and iii) the state-of-the-art catalytic systems reported to date. More importantly, Pt-Mo/NC$_{0.07}$ exhibited stable performance in a 50 h time-on-stream without any apparent activity or selectivity deterioration, thus demonstrating excellent structural robustness. The operando DB-FTIR coupling with DFT modeling jointly demonstrated the superior catalytic performance of the low-nuclearity Pt-Mo ensembles rooted at the unique $Pt_3Mo_1N_3$ configuration, over which the adsorption and dissociation of HCOOH were thermodynamically favorable. Furthermore, the synergistic catalysis between the Pt and

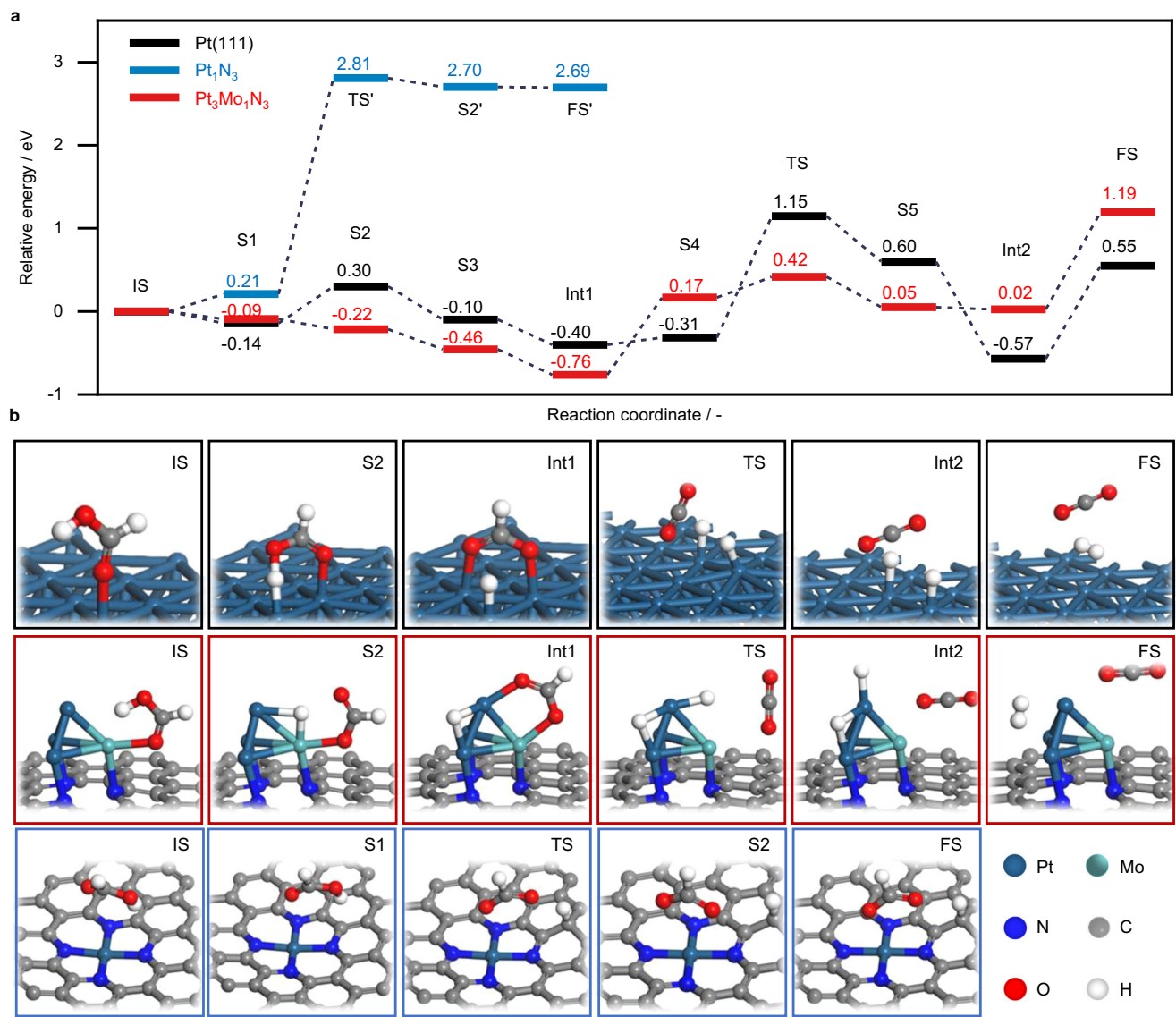

**Fig. 5 | Mechanistic insights from DFT. a** Energy profiles for the decomposition of formic acid on the different model catalysts. **b** Side view of DFT-optimized key adsorption configurations.

Mo sites provided an alternative path for lowering the energy barriers for the consecutive HCOOH dissociations, which were otherwise most energy-demanding on the nanoparticle or single-atom Pt catalysts. To develop a practical HCOOH-to-H₂ technology, it is still imperative to design more efficient catalytic materials based on non-noble metals which can operate ideally at ambience conditions. In this scenario, the developed defect-driven nanostructuring approach may offer a new opportunity to tackle this challenge through the rational design of more sophisticated low-nuclearity heteroatom ensembles.

## Methods
### Catalyst preparation
An oxidative polymerization method was applied to prepare poly-anilines (PAni) following our previously reported recipe.[53,65] First, 45.8 cm³ aniline (≥99.5 wt.%, Aladdin) was dissolved in 400 cm³ aqueous solution of HCl (1.25 mol dm⁻³, 37%, AR, Xi Long Scientific) and the mixture were stirred vigorously. Meanwhile, 114.4 g ammonium persulfate (≥98 wt.%, Aladdin) was dissolved in 200 cm³ deionized water. The above two solutions were transferred into a refrigerator (preset to 277 K) for 1 h. Afterwards, the ammonium persulfate solution was added to the aniline solution with vigorously stirring in order to initiate

the polymerization process. After continuous stirring for an additional 5 min, the mixture was left static overnight. Note that, the above procedures were conducted at room temperature although the reaction heat could build up at the initial stage of the polymerization. The as-obtained solids were then washed with plenty deionized water (4 L) and dried at 373 K overnight to generate PAni. To yield N-doped carbons (NC), the PAni was further carbonized at different temperatures of 1073–1673 K (ramping rate 5 K min⁻¹, 1 h). The resultant supports were coined as $NC_x$, $x = 0.004$-$0.13$, wherein $x$ denoted the molar N:C ratios.

For the preparation of the bimetallic catalysts, the metal precursors were introduced into the NC carriers by a sequential incipient impregnation method. First, an aqueous solution of $H_{24}Mo_7N_6O_{24}·4H_2O$ (99%, Macklin) was added into $NC_x$ (20–40 meshes) dropwise, swiftly stirred with a glass rod, and dried in an oven at 353 K for 12 h. Afterward, the solids were placed in a tubular furnace and then heated to 773 K (ramping rate 10 K min⁻¹, 5 h, Ar atmosphere). After cooling to room temperature, the samples were impregnated with an aqueous solution of $H_2PtCl_6·6H_2O$ (99.9% Pt, Sigma-Aldrich) in a similar manner, dried in an oven at 353 K for 12 h, placed in a tubular furnace and heated to 773 K (ramping rate 5 K min⁻¹, 4 h, Ar

atmosphere). After cooling to room temperature, the solids were further treated with a mixture of 20 vol.% $CH_4$/80 vol.% $H_2$ at 573 K (ramping rate 5 K min$^{-1}$, 1 h) and 973 K (ramping rate 1 K min$^{-1}$, 2 h). The derived catalysts were denoted as Pt-Mo/NC$_x$. The preparation of the mono-metallic catalysts also followed the incipient impregnation of the respective metal precursors and the subsequent multi-heating programs that were similar to the procedures of the bimetallic samples. These catalysts were denoted as Pt/NC$_{0.07}$ and Mo/NC$_x$. An additional catalyst Pt/NC$_{0.02}$ was prepared in a similar manner but with a lower reduction temperature of 573 K. The nominal Pt and Mo contents of all the designed catalysts were 0.5 and 5.0 wt.%, respectively. Commercial $Mo_2C$ was purchased from Macklin (99.95%).

## Catalyst characterization

Nitrogen sorption at 77 K was performed on a Quantachrome Autosorb-1 instrument. Before the analysis, the solids were sufficiently degassed at 573 K for 3 h. The specific surface area was calculated by the five-point Brunauer-Emmett-Teller (BET) method and the total pore volume was calculated at the $p/p_O = 0.97$. before the measurements. The C, H, and N contents of the solids were measured by infrared spectroscopy using a LECO TruSpec Micro combustion furnace. The Pt and Mo contents in the catalysts were measured by ICP-OES on a Perkin-Elmer Optima 7300DV. To help fully digest the solids, a mixture of $H_2O_2$ and aqua regia with the assistance of microwave irradiation on Anton Paar Multiwave 3000 was adopted so that no solid residual was left before the measurement. PXRD analysis was performed on an X'Pert 3, PANalytical X-ray diffractometer using Cu $K_\alpha$ radiation in a scanning angle ($2\theta$) range of 10–90° with a scanning speed of 0.2° min$^{-1}$. The tube voltage and the current were set at 40 kV and 40 mA, respectively. Raman spectra were acquired on a confocal laser micro-Raman spectrometer (HORIBA, Lab RAM HR Evolution, wavelength 532 nm, 60 s transits per sample, resolution 1 cm$^{-1}$). Transmission electron microscopy (TEM) measurement was conducted on a JEM-2100 transmission electron microscope applying an accelerating voltage of 200 kV. The solid powders were dispersed in ethanol with ultrasonication and then the specimen was transferred onto a carbon film supported on a copper grid by dropping a droplet suspension for the measurement. The HAADF-STEM images were collected on a JEOL-ARM200F electron microscope. X-ray photoelectron spectroscopy (XPS) measurements were conducted on a Thermo ESCALAB 250Xi spectrometer using a 15 kV Al $K_\alpha$ X-ray source as a radiation source. The C-1$s$ peak at a binding energy of 285 eV was used as the reference since the doping of N into the carbon framework was known to upshift the binding energies[66,67]. X-ray absorption fine structure (XAFS) data of Pt $L_3$-edge and Mo $K$-edge were collected at beamline BL14W1 of Shanghai Synchrotron Radiation Facility (SSRF). The data were recorded in the fluorescence mode equipped with Electro-Lyte detector. The original EXAFS data were analyzed by the Demeter software package. Fourier transformation was applied to process the $k^3$-weighted raw data. The theoretical scattering amplitude and phase-shift functions of all the paths for fitting the EXAFS data were calculated with FEFF6 code.

The dual-beam Fourier transform infrared spectra (DB-FTIR) were obtained on home-made equipment equipped with MCT detectors in the range of 4000–400 cm$^{-1}$ with an optical resolution of 4 cm$^{-1}$. The catalysts were mixed homogeneously with KBr at the mass ratio of 1:20, and then pressed into thin wafers (ca. 50 mg per 1.5 cm$^2$), then placed in the sample beam of the dual-beam IR cell. The reference beam in the absence of catalysts was taken as the reference spectrum. The wafers were pretreated at 673 K for 1 h under a high vacuum ($5 \times 10^{-3}$ Pa). Then catalyst reference spectra were collected after the cell cooled down to room temperature. Analytic pure formic acid vapor was introduced into the IR cell for 5 min. The catalysts were purged in flowing $N_2$ gas for 30 min and then desorbed by temperature ramping in the range of 303–473 K. The series spectra of adsorption spectra (sample beam) and gas reference spectra (reference beam) were collected simultaneously as time-resolved spectra during the whole desorption process. For CO adsorption at liquid nitrogen temperature (77 K), the pretreatment and adsorption procedures were the same as those mentioned above. After pretreatment, 20 cm$^3$ 1% CO/$N_2$ was introduced into IR cell, and series spectra were collected. The sophisticated spectra of surface group vibrations were calculated by the subtraction of the catalyst reference spectra and gas reference spectra from adsorption spectra:

Surface group vibrations spectra = Adsorption spectra − Catalyst reference spectra − Gas reference spectra.

## Catalytic testing

The catalytic performance of the developed catalysts in the continuous gas-phase decomposition of formic acid was conducted in a home-made fixed-bed quartz reaction at ambience pressure (Supplementary Fig. 15). The catalyst pellets (20–40 meshes, $W_{cat} = 200$ mg) were packed between glass wools in the middle of a quartz reactor (300 mm length, 10 mm internal diameter), which was vertically placed in an oven equipped with a temperature controller. Before the reaction, the catalysts were first heated to 573 K (10 K min$^{-1}$, $N_2$ 20 cm$^3$ min$^{-1}$) and stayed for 1 h. Then the catalysts were reduced in situ by switching the gas to pure $H_2$ (99.99 vol.%, 20 cm$^3$ min$^{-1}$, 1 h). The catalyst bed was cooled to the desired temperatures and then the reaction was initiated by admitting the feeds comprising 10 wt.% HCOOH in deionized water at a speed of 10 cm$^3$ min$^{-1}$, and a stream of 10 vol.% $N_2$/He or pure He as the carrier gas (40 cm$^3$ min$^{-1}$, $N_2$ used as an internal standard for the peak calibration in the gas chromatograph). The content of formic acid in the feed was 4.15% in mole, and the corresponding GHSV was 15,000 cm$^3$ g$^{-1}$ h$^{-1}$. A cold trap was placed at the outlet of the reactor to collect the cooled formic acid and the water, and the rest gaseous products were analyzed online by an Agilent 7890B chromatograph equipped with a TDX-01 column and a thermal conductivity detector. $CO_2$ and CO were the only products detected. The conversion ($X$) and $CO_2$ selectivity ($S_{CO2}$) were calculated by Eqs. 1 and 2, respectively,

$$X (\%) = (F_{CO2,out} + F_{CO,out})/F_{FA,in} \times 100 \qquad (1)$$

$$S_{CO2} (\%) = F_{CO2,out}/(F_{CO2,out} + F_{CO,out}) \times 100 \qquad (2)$$

wherein, $F_{CO2,out}$ and $F_{CO,out}$ were the outlet flow rates (mol min$^{-1}$) of $CO_2$ and CO, respectively, and $F_{FA,in}$ was the inlet flow rates (mol min$^{-1}$) of formic acid. Trace CO was quantified by referring the standard gases (10 and 50 ppm CO/He).

The turnover frequency (TOF) based on the total content of Pt was calculated by Eq. 3,

$$TOF = M_{HCOOH}/(w_{Pt} \times w_{cat.}/M_{Pt}) \qquad (3)$$

wherein, $M_{HCOOH}$ was the mol of HCOOH converted per second at <20% conversion, $w_{Pt}$ was the Pt content determined by the ICP analysis, $w_{cat.}$ was the weight of the catalyst loaded in the reactor, and $M_{Pt}$ was the molecular weight of Pt. Herein, the activity was normalized by the total number of Pt atoms because the CO probe molecule can hardly be chemisorbed on the Pt-based catalysts even at 77 K (Supplementary Fig. 29). Noted that, the low-nuclearity nature of Pt-Mo/NC$_{0.07}$ might indicate close TOF values by the different normalization methods.

## Computational method

The DFT calculations were performed using the Vienna Ab initio Simulation Package (VASP)[68]. The calculation of the electronic exchange-correlation term was described by the generalized gradient

approximation method with the Perdew-Burke-Ernzerhof functional. The interaction between the atomic nucleus and the valence electron was described by the conjugated projection wave pseudopotential. The cutoff energy for the plane wave basis set was fixed at 450 eV. In order to achieve convergence, the Gaussian smearing method was used with a bandwidth of 0.05 eV. K-points mesh was optimized to be $3 \times 3 \times 1$ Γ-centered. The convergence criteria of structure optimization and energy calculation were as follows: energy convergence of $1.0 \times 10^{-7}$ eV, and atomic force convergence of $0.05$ eV Å$^{-1}$. We employed the climbing image nudged elastic band method to compute the transition states[69]. The DFT calculations used a model with a $12 \times 12$ Å unit cell and constructed a two-layer graphene structure. The upper layer of the graphene was relaxed, while the lower layer was fixed. Different numbers of pyridinic and pyrrolic nitrogen defects with the introduction of different single atoms or clusters were constructed on the upper layer.

The adsorption energy of reactant was calculated through Eq. 4,

$$E_{ad} = E_{adsorption\ state} - E_{HCOOH} - E_{catalyst} \qquad (4)$$

The dissociation energy of reactant was calculated through Eq. 5,

$$E_d = E_{dissociation\ state} - E_{reactant} - E_{slab} \qquad (5)$$

The formation energy of catalyst was calculated through Eq. 6,

$$E_f = E_{catalyst} - E_{cluster} - E_{slab} \qquad (6)$$

The binding energy of catalyst was calculated through Eq. 7,

$$E_b = (E_{catalyst} - E_{slab} - NxE_{atom})/N \qquad (7)$$

## Data availability
Data presented in the main figures of the manuscript are publicly available through the Figshare repository (https://doi.org/10.6084/m9.figshare.23896251). Further data supporting the findings of this study are available in the supplementary Information. All other data are available from the authors upon request. Source data are provided with this paper.

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

## Acknowledgements

We thank the National Key Research and Development Program of China (2021YFA1501801, 2022YFA1504402), the National Natural Science Foundation of China (No. 22002151), the Strategic Priority Research

Program of the Chinese Academy of Sciences (No. XDA29050300, XDA21020201 and XDA21020300), and Zhejiang Provincial Key R&D Project (2023C01211), and Zhejiang Normal University for providing the financial support (ZZ323205020521005039, KYJ51020910 and YS304320036). We thank Profs. C.W. Yang, Z.P. Chen, and Dr. H.Z. Guo for their helpful discussions.

## Author contributions

Y.D., R.L., and H.Z. conceived and coordinated the study. L.G., R.L., J.Z., and J.L. wrote the article with input from all other co-authors. L.G., S.W., X.M., L.Y., and Z.Z. synthesized the catalysts, contributed to their characterizations, and conducted the catalytic tests. S.Y. and J.L. performed the infrared spectroscopic analyses. P.Q. performed the XAS. K.Z., S.W., and J.Z. conducted DFT calculations and fitting of the X-ray absorption spectra. All authors contributed to the writing of the manuscript.

## Competing interests

The authors declare no competing interest.
