## [Peer Review File · Nature Communications]

REVIEWER COMMENTS

Reviewer #1 (Remarks to the Author):

In the manuscript "Defect-driven nanostructuring of low-nuclearity Pt-Mo ensembles for continuous gas-phase formic acid dehydrogenation" Guo et al; report a defect-driven nanostructuring strategy through combining defect engineering of nitrogen-doped carbons and sequential metal depositions to prepare a series of Pt and Mo ensembles ranging from single atoms to sub-nanoclusters. They propose that low-nuclearity ensembles with unique Pt₃Mo₁N₃ configurations deliver CO-free hydrogen at full conversion with unexpectedly high activity and remarkable stability in continuous gas phase formic acid decomposition.

In their short review, they discuss the merits and drawbacks of biomass-to-biochar catalyst conversion and the most relevant factors to consider in the synthesis stage for enhancing catalytic activities.

The manuscript is interesting, but the results found have been reported previously in the literature, so I do not find them to be of sufficient quality for the journal in which they are proposed for publication. For that main reason and the following points, I propose that it be rejected for publication in Nature Communications.

1. Although the introduction is well written, a real comparison with the most recent results obtained for the dehydrogenation of formic acid in liquid media is lacking.
2. The platinum particle distributions in Figure 2 are not faithful to reality. In the HAADF micrographs, platinum particles larger than 2-3 nm are observed (for example, in figure 2d), which are not contemplated in the particle distribution.
3. Assuming that the platinum atomic radius is 175 pm, how is it possible that particle distributions, with an average size smaller than the size of the platinum atomic radius, have been found (figure 2a and c).
4. Figure s4 shows the diffractogram of the obtained samples. A peak associated with the metallic platinum species is clearly visible, which means that the average crystal size is larger than 5 nm (the stopping limit of the technique).
5. However, the authors propose that the average size of the metallic particles, calculated by HR-TEM, figure 2e, is 0.66 nm. How do the authors explain this controversy?

6. 4. The authors claim that the post-reaction sample shows no apparent modifications. However, there is an accentuation of the peak (110) of amorphous carbon, which could indicate a partial graphitisation of the carbonaceous support. On the one hand, the authors should show the XRD diffractogram from 10-90° 2 Theta, to observe the first carbon diffraction peak, and be able to compare them before and after the reaction, correlating these modifications when calculating the La and Lc parameters, and the packing factor, with the possible presence of carbonaceous leftover species.

For this reason, the work might not be published in Nature communications.

Reviewer #2 (Remarks to the Author):

What are the noteworthy results?

The authors reported a defect-driven nanostructuring strategy through combining defect engineering of nitrogen-doped carbons and sequential metal depositions to prepare a series of Pt and Mo ensembles ranging from single atoms to sub-nanoclusters. When applied in continuous gas-phase decomposition of formic acid, the low-nuclearity ensembles with unique Pt₃Mo₁N₃ configuration deliver CO-free hydrogen at full conversion with high activity and good stability.

Will the work be of significance to the field and related fields? How does it compare to the established literature? If the work is not original, please provide relevant references.

The reported work is interesting because it can produce CO-free hydrogen over 50 hours from formic acid. However, to have a significant impact on hydrogen applications using formic acid as the energy carrier, the authors need to consider the following:

(1) Operating temperature: Even though the volumetric energy density of formic acid is higher than hydrogen gas, it is still very low compared to other liquid hydrogen carriers. The significant advantage of formic acid as a hydrogen carrier over other liquid hydrogen carriers (e.g., methanol or ethanol) is that it can reform at room temperature. As the operating temperature increases beyond the room temperature, the advantage of using formic acid as a hydrogen carrier diminishes. In order to have a significant impact, it is important that the catalyst needs to give both high activity and good stability at the ambient conditions (e.g., 25 °C and 1 atm) for the formic acid decomposition reaction. The reported

Pt-Mo/NC catalyst is interesting, but it still requires a very high temperature of 100 °C to provide good performance. The real practical technological challenge for the formic acid decomposition reaction in hydrogen application is developing non-noble metal catalysts (or ultra-low precious metal catalysts) that can deliver CO-free hydrogen at ambient conditions over a long time.

(2) CO-free hydrogen: The gas stream from the reformer has to be free from CO gas (<10 ppm), or the catalytic performance of the fuel cells will be degraded significantly. Based on the given experimental data, it is hard to determine whether there were any ppm levels of CO gas or not. The author should provide information about CO gas concentration in ppm levels.

Does the work support the conclusions and claims, or is additional evidence needed?

(1) The authors stated that “Mo atoms are first strongly coordinated by the N defects followed by the deposition of Pt, wherein controlled aggregation of Pt is realized by simply reducing the numbers of nitrogen defects.” The authors need to clearly articulate how the N defects can coordinate with Mo atoms, and how this coordination chemistry can be affected by the concentration of N doping. Furthermore, there are no clear experimental data and analysis to explain how these N defect-controlled Mo atoms interact with Pt atoms and control the final ensemble size of Pt. What is the exact role of Mo atoms in controlling the ensemble size of Pt in the overall synthesis process?

(2) Based on the performance data, Pt-Mo/NC0.07 shows an improved activity compared to Pt/NC0.07. This improved activity could be originated from the ensemble size effect and/or the presence of Mo atoms. As the ensemble size changes and Mo atoms are introduced to Pt clusters, both their physicochemical and electronic properties will change and affect the catalytic performances. The authors provided DFT calculation data to explain the ensemble size effect and role of Mo atoms. However, their DFT data do not sufficiently address the size effect and the nature of the synergistic catalysis of the Pt-Mo ensembles. Could they prepare the samples with and without Mo while fixing the ensemble size? This comparison would allow them to isolate the Mo effect for example.

(3) The authors stated that “DB-FTIR study evidenced the different kinetic fingerprints of formic acid molecules with the distinct mono- and bimetallic sites as illustrated in Fig. S16.” However, this reviewer cannot see how the DB-FTIR data shown in Figure 4 can lead to Figure S16. The DB-FTIR data do not reveal the distinct mono- and bimetallic sites. Instead, they simply show the different fingerprints of formic acid and its derivatives as the function of temperature. Based on the DB-FTIR data, it is very hard to make any conclusive statements about the degree of interaction between the specific sites of the catalyst and formic acid (and its derivatives). For example, the authors stated that “On the representative single-atom Pt sites, strong adsorption of formic acid was observed, but the deprotonation was much difficult.” How can the authors prove that the deprotonation (of first H from O-H or second H from C-H?) was harder over the single-atom Pt sites than other catalysts?

(4) The authors assigned the adsorption bands of the DB-FTIR data at 1718 and 1595 cm⁻¹ as the C-O vibration of the molecularly adsorbed HCOOH and the O-C-O vibration due to the adsorbed formate

species. If this is true, as the temperature increases the peak intensity of HCOOH_{ad} should decrease while the peak intensity of HCOO_{ad} should increase followed by the decrease. Do the DB-FTIR data show these trends? In general, the interpretation of the DB-FTIR data needs to be significantly improved and better match them to the predicted DFT-derived reaction mechanisms.

(5) Based on operando DB-FTIR data (Figure 4 c and d), it seems that both Mo/NC0.07 and Pt-Mo/NC0.07 show very similar spectra. However, their activities are very different (Pt-Mo/NC0.07 shows the best performance while Mo/NC0.07 shows the worst performance). To explain this, the authors speculated that the abstraction of another H atom might be more energy-demanding than the dissociation of HCOOH. To support its speculation, the authors should show the energy profiles for the formic acid decomposition over the Mo clusters or Mo single atoms (whichever best represents the Mo/NC0.07 sample).

(6) For the DFT energy profile of Pt₃Mo₁N₃, the second H (from C-H bond) is deprotonated by chemisorbing to the nearest Pt site. However, in order for this H transfer to occur, this H must first move closer to the nearest Pt site. Because this H (from C-H) is located too far away from the nearest Pt site, this reviewer is not entirely convinced that such a transfer reaction can occur with the energy downhill. This should be a very unfavorable reaction because adsorbed HCOO needs to be stretched out to the nearest Pt site in an extreme degree.

Reviewer #3 (Remarks to the Author):

The authors reported defect-driven nanostructuring strategy for the preparation of well-dispersed bimetallic Pt-Mo species on nitrogen-doped carbon, which have shown good catalytic activity in gas-phase formic acid dehydrogenation.

Although the investigation comprises a systematic study of the material properties, catalytic evaluation, and mechanism, the following issues should be addressed before consideration for publication.

1. From the XPS spectra, the authors suggested the presence of Pt-N interactions. However, CH₄ was used during the reduction process, which could lead to the formation of metal carbides (Catal. Sci. Technol., 2020, 10, 6790–6799). It is thus important to show whether Pt-C or Mo₂C was formed.
2. From the TEM studies, it was evident that some of the impregnated metal atoms were present as metal nanoparticles or metal clusters. However, surface sensitive techniques such as XPS did not show any Pt-Pt bond or Mo-Mo in the case of isolated impregnations (Pt/NC0.07 and Mo/NC0.07). These observations contradict the TEM studies, so the authors should discuss this anomaly.
3. Likewise, if the isolated impregnations do not show cluster formation, how can the bimetallic systems give rise to Pt-Pt clusters?

4. The cluster sizes of the Pt-Mo/NC_{0.07} (1-4 atoms) were estimated 10-30 times smaller than those of Pt-Mo/NC_{0.02} (13-133 atoms), but why did the TEM images of both materials seem to show a high degree of agglomerated cluster?

5. The FTIR studies suggested that Mo/NC may be a major active species (or partially carbonized MoxCy species) since both the Mo/NC_{0.07} and Pt-Mo/NC_{0.07} showed a similar dehydration pathway associated at elevated temperatures. This was contradicting with the plot "Fig. 3a" which shows 100% CO₂ selectivity for Mo/NC_{0.07}, while its FTIR shows minor quantity of CO-related peaks at 2190 cm⁻¹.

6. The authors stated that "gas-phase dehydrogenation is attractive considering the mild exothermicity and ease of catalyst separation as compared with the endothermic liquid-phase dehydrogenation (-15 vs. 29 kJ mol⁻¹)"; however, vapor-phase dehydrogenation is operated at relatively higher temperatures. In the case of dilute feed (10% FA in water), it also generated unwanted steam, which is an endothermic process. Since 90% of water is being used in the feed, a significant amount of energy will be lost for vaporization. Have the authors proposed any heat recovery mechanism? If an inert gas such as helium is used as a carrier gas, are additional separation steps needed for hydrogen purification?

Minor comments:

7. The authors need to clarify whether metal loading is in weight percentage or atomic percentage. Text and table values were given in different units.

8. The references about the reviews for formic acid dehydrogenation should be updated.

Response to Reviewers

Comments in blue - Replies in black - Actions in **bold**, citation – in *italic*

Reviewer #1

In the manuscript "Defect-driven nanostructuring of low-nuclearity Pt-Mo ensembles for continuous gas-phase formic acid dehydrogenation" Guo et al; report a defect-driven nanostructuring strategy through combining defect engineering of nitrogen-doped carbons and sequential metal depositions to prepare a series of Pt and Mo ensembles ranging from single atoms to sub-nanoclusters. They propose that low-nuclearity ensembles with unique Pt₃Mo₁N₃ configurations deliver CO-free hydrogen at full conversion with unexpectedly high activity and remarkable stability in continuous gas phase formic acid decomposition.

In their short review, they discuss the merits and drawbacks of biomass-to-biochar catalyst conversion and the most relevant factors to consider in the synthesis stage for enhancing catalytic activities.

The manuscript is interesting, but the results found have been reported previously in the literature, so I do not find them to be of sufficient quality for the journal in which they are proposed for publication. For that main reason and the following points, I propose that it be rejected for publication in Nature Communications.

We sincerely thank the Reviewer 1 for his/her interest and his/her careful assessment of this work. However, we respectfully disagree with the Reviewer on that "the results found have been reported previously in the literature". The novel aspects of our work clearly distinguishing from the previous findings are briefly summarized as follows:

i) **Novel material design strategy.** As stated in the Introduction of the previous manuscript, the rational design of low-nuclearity bimetallic catalysts is still a great challenge. The currently established approaches are mainly restricted to atomic layer deposition methods or confined pyrolysis of MOF/COF precursors containing hetero-atoms. Contrasting to the previous methods that generally lack control of the ensemble sizes, delicate tuning of the ensemble sizes in the subnano regime (from dual atoms to ca. 0.66 nm) is realized in our work. Our key idea is that, i) Mo atoms are introduced in the first deposition and strongly coordinated by the N defects after thermal treatment, ii) Pt atoms are sequentially introduced in the second deposition, iii) controlled aggregation of Pt is realized by reducing the numbers of nitrogen defects which act as the coordination sites for both Pt and Mo atoms. To our knowledge, it is the first time that defect-driven nanostructuring strategy in combination with sequential deposition of different metals is proposed for the design of bimetallic ensembles ranging from atomic dispersions to the subnano regime. Compared with the reported methods which require advanced facilities or preparation of complex precursors, our approach is more straightforward to practical applications.

ii) **Outstanding catalytic performance of new Pt-Mo/NC catalytic system.** It is the first time that bimetallic Pt-Mo catalysts were proposed for the dehydrogenation of HCOOH. In particular, the low-nuclearity Pt-Mo/NC_{0.07} catalyst exhibit outstanding performance in the continuous gas-phase HCOOH dehydrogenation as compared with the earlier reported analogues. Regarding the utilization efficiency of precious metals, Pt-Mo/NC_{0.07} outperformed the other gas-phase dehydrogenation catalysts by ca. one order of magnitude in the total metal-based reaction rates, and reached close or even better values than those of the state-of-the-art liquid-phase

dehydrogenation catalysts. (The Reviewer is kindly asked to refer to **Fig. 3d** and supplementary **Table 1**).

iii) **Firstly reported Pt-Mo synergy.** We clearly demonstrated that the Pt-Mo ensembles in Pt-Mo/NC_{0.07} superior catalytic performance in HCOOH dehydrogenation than any of the monometallic catalysts, including single Mo, Pt, and Pt clusters and nanoparticles. The superior catalytic performance of Pt-Mo/NC_{0.07} is further rationalized by combining *operando* DB-FTIR and DFT simulations, demonstrating that the unique Pt₃Mo₁N₃ configuration can create a new reaction path with lower energy barrier for HCOOH dissociation by the cooperative catalysis of both Pt and Mo sites.

In fact, these innovative points have been recognized and praised by both Reviewers 2 and 3. We trust that the novelty of this work can be valuable and of high interest to the broad readers in material science, physical chemistry, and heterogeneous catalysis. To further address the Reviewer's questions/concerns, a detailed point-to-point reply has been provided in the following:

1. Although the introduction is well written, a real comparison with the most recent results obtained for the dehydrogenation of formic acid in liquid media is lacking.

Thank you for your praise and the comment. In the previous manuscript, we briefly compared the pros and cons of the liquid- and gas-phase processes for the dehydrogenation of formic acid, and the state-of-the-art liquid-phase dehydrogenation catalysts were also introduced. For instance, we have stated that '*Liquid-phase dehydrogenation is typically performed with the presence of homogeneous (e.g., Ru, Ir, and Fe-based organometallic complexes)²⁵⁻²⁷ or Pd-based heterogeneous catalysts^{28,29}. Both systems can afford high catalytic efficiency with the turnover frequency (TOF) reaching 0.31-5.97 s⁻¹ with typically negligible CO formation at low operation temperatures³⁰. Nonetheless, the tedious recycling of the expensive organometallic complexes and the frequently reported deactivation of the heterogeneous catalysts due to the site blockage induced by trace CO prevent a prospective industrial process³¹.*' (see **Page 4, lines 19-23; Page 5, lines 1-2**)

Since the current work focused mainly on the gas-phase formic acid dehydrogenation, more attentions were paid on the introduction of gas-phase dehydrogenation catalysts which were drawn for the direct comparison on their catalytic performance. To address the Reviewer's comment, **we have now accommodated the catalytic performances of the liquid-phase dehydrogenation catalysts from the most recent literatures for comparison with that of our developed catalyst in the revised piece (see supplementary Table 1).**

Table S1. Comparison on the performance between Pt-Mo/NC_{0.07} and the literature reported catalysts in the gas-phase dehydrogenation of formic acid.

Catalysts	Precious-metal	Feed	50% conversion		TOF	rate ^b	E _a	Stability			refs.
	wt.%		T/K	S _{H₂} /-% ^a	at 373 K/s ⁻¹	s ⁻¹		/kJ·mol ⁻¹	r _d /h ^{-1e}	T/K	
Pd-K/C	1.0	2.0% HCOOH-He	345	99	-	-	97	-	-	-	[1]
Pd/N-CNFs	1.0	1.9% HCOOH-He	427	-97.6	0.21(398K)	0.063	-	-	-	-	[2]
Pd/N-PCN	1.0	1.9% HCOOH-He	422	-95.6	0.27(398K)	0.049	-	-0.33	448	30	[2]
Pd/Mel/C	1.0	2.5% HCOOH-Ar	528	≥98	-	0.079(393K)	32	-1.5	573	6	[3]
Pt/N-CNFs	1.0	2.0% HCOOH-He	445	99.5	0.088	-	43-53	-	-	-	[4]
Pt/N-CNFs	0.3	1.8% HCOOH-He	469	99.6	0.25(398K)	0.038(398K)	-	-0.14	448	42	[5]
Ir/AC	2.0	5.0% HCOOH-Ar	383	100	0.096	0.025	-	-	-	-	[6]
Pt/AC	2.0	5.0% HCOOH-Ar	423	98	0.064	0.015	-	-	-	-	[6]
Pt/N-graphene	1.0	5.0% HCOOH-He	443	98	0.42(398K)	0.025	-	0	423	5	[7]
Au/Al ₂ O ₃	2.5	1.9% HCOOH-He	456	99.6	-	0.0072	57	-0.43	423	16	[8]
Mo ₂ C/AC	0	5.5% HCOOH-N ₂	423	95	0.066	-	-	0	453-493	12	[9]
Mo ₂ C-Co/AC	0	-	445	99.5	0.088	-	-	-	-	-	[10]
Pt-Mo/NC _{0.07}	0.48	4.15% HCOOH-H ₂ O-N ₂ -He	360	100	0.31	0.31	54	0	373	50	*
					0.62(388K)	0.62(388K)					*

Note: ^aH₂-selectivity. ^brate in mol of HCOOH converted per mol of precious metals per second. ^cDeactivation rate defined by the change in HCOOH conversion in percentage per hour during stability tests. *This work.

Supplementary Table 1. Comparison on the performance between Pt-Mo/NC_{0.07} and the literature reported solid catalysts in the gas- and liquid-phase dehydrogenation of formic acid.

Catalysts	Precious metal wt. %	Feed	50% conversion		TOF at 373 K / s ⁻¹	rate ^b s ⁻¹	E _a / kJ mol ⁻¹	Stability			refs.
			T / K	S _{H2} / % ^a				r _d / h ^{-1 c}	T /K	Duration / h	
Gas-phase dehydrogenation											
Pd-K/C	1.0	2.0%FA-He	345	99	-	-	97	-	-	-	[1]
Pd/N-CNFs	1.0	1.9% FA-He	427	~97.6	0.21(398K)	0.063	-	-	-	-	[2]
Pd/N-PCN	1.0	1.9%FA-He	422	~95.6	0.27(398K)	0.049	-	-0.33	448	30	[2]
Pd/Mel/C	1.0	2.5%FA-Ar	528	>98	-	0.079(393K)	32	-1.5	573	6	[3]
Pt/N-CNFs	1.0	2.0%FA-He	445	99.5	0.088	-	43-53	-	-	-	[4]
Pt/N-CNFs	0.3	1.8%FA-He	469	99.6	0.25(398K)	0.038(398K)	-	-0.14	448	42	[5]
Ir/AC	2.0	5.0%FA-Ar	383	100	0.096	0.025	-	-	-	-	[6]
Pt/AC	2.0	5.0%FA-Ar	423	98	0.064	0.015	-	-	-	-	[6]
Pt/N-graphene	1.0	5.0% FA-He	443	98	0.42(398K)	0.025	-	0	423	5	[7]
Au/Al ₂ O ₃	2.5	1.9%FA-He	456	99.6	-	0.0072	57	-0.43	423	16	[8]
Mo ₂ C/AC	0	5.5%FA-N ₂	423	95	0.066	-	-	0	453-493	12	[9]
Mo ₂ C-Co/AC	0	-	445	99.5	0.088	-	-	-	-	-	[10]
Pt-Mo/NC _{0.07}	0.48	4.15%FA-H ₂ O-N ₂ -He	360	>99.9	0.31	0.31	54	0	373	50	*
					0.62(388K)	0.62(388K)					*
Liquid-phase dehydrogenation											
Catalysts	Precious metal wt. %	Feed	T / K	CO / ppm	TOF / s ⁻¹	rate ^b s ⁻¹	E _a / kJ mol ⁻¹				refs.
Pd/S-1-in-K	0.64	FA-SF-H ₂ O	323	<10	0.84	0.91	39				[11]
in situ -Pd/MS	4.6	FA-SF-H ₂ O	333	n.d.	2.53	4.44	31				[12]
Pd ₁ Au ₁ /30-LA	3.4Au, 1.8Pd	FA-SF-H ₂ O	333	n.d.	2.32	-	39				[13]
Pd@CN900K	9.3	FA-SF-H ₂ O	333	<10	4.00	-	47				[14]
Pd/BNF-C	9.1	FA-SF-H ₂ O	313	n.d.	1.06	-	36				[15]

(Co-N)6.8@NC	6.8	FA-propylene carbonate	393	n.d.	-	0.012	47	[16]
PdNi-WO _x /KIT-6-NH ₂	9.98Pd	FA-H ₂ O	323	n.d.	1.22	-	35	[17]
HCOONa-Pd@ANI/C	3.74	FA-SF-H ₂ O	303	n.d.	-	0.58	-	[18]
Pd-WO _x /(P)NPCC	14.3Pd	FA-SF-H ₂ O	323	n.d.	1.70	-	36	[19]
γ-Mo ₂ N/0.2NK-C	26Mo	FA-H ₂ O	360	n.d.	-	1.4x10 ⁻⁵	-	[20]

Note: ^aH₂ selectivity. ^brate in mol of HCOOH converted per mol of precious metals per second. ^cDeactivation rate defined by the change in HCOOH conversion in percentage per hour during stability tests. *This work. FA: formic acid, SF: sodium formate. n.d.: not detected.

2. The platinum particle distributions in Figure 2 are not faithful to reality. In the HAADF micrographs, platinum particles larger than 2-3 nm are observed (for example, in figure 2d), which are not contemplated in the particle distribution.

We thanked the Reviewer 2 for pointing out the mistake which we deeply apologize. The X scale in **Fig. 2d** was indeed not fully displayed in the previous manuscript. It should be 0-1.5 nm, instead of 0-1.0 nm. This mistake aroused due to the very low frequency of the particles larger than 1 nm, and consequently the X scale was unintentionally set up to 1 nm during the potting of this figure. **This mistake has now been corrected in the revision.** The Reviewer is also kindly reminded that each histogram was made based on >500 entities counted and the biggest particle size in **Fig. 2d** is actually about 1.42 nm as displayed in the zoom figure (see **Fig. R1**).

Fig. 2. Characterization of the key catalysts. a-e, HAADF-STEM images with the particle size distributions. **f,g**, Normalized XANES spectra with the Pt and Mo foils references. **h,i**, EXAFS and the fitting results.

Fig. 2. Characterization of the key catalysts. a-e, HAADF-STEM images with the particle size distributions. **f,g**, Normalized XANES spectra with the Pt and Mo foils references. **h,i**, EXAFS and the fitting results.

Fig. R1. The enlarged image from **Fig. 2d**, showing the largest particle size of about 1.42 nm for Pt-Mo/NC_{0.07}.

3. Assuming that the platinum atomic radius is 175 pm, how is it possible that particle distributions, with an average size smaller than the size of the platinum atomic radius, have been found (figure 2a and c).

Thank you for the comment. We should kindly remind the Reviewer that these images in **Fig. 2** were acquired by HAADF-STEM technique. In principle, STEM imaging relies on the interactions of the atomic nucleus with the elastically scattered electrons. When sizes of the detecting objects decrease from large metal nanoparticles to single atoms, the white spots in the HAADF-STEM images are not the reflections of the metal atoms, but the atomic nucleus instead. In the case of Pt, the radius of Pt²⁺ is much smaller than Pt atoms (80 vs. 138 pm). In addition, the outline of the white spots might also be influenced by the resolution of the recording camera. For these reasons, the sizes of atomically dispersed metals derived from the STEM is not exactly the same as those of the theoretical ones. Therefore, for single-atom catalysts, STEM imaging can provide valuable information based on the agglomeration of metal species, but cannot be used to determine the atomic size.

4. Figure s4 shows the diffractogram of the obtained samples. A peak associated with the metallic platinum species is clearly visible, which means that the average crystal size is larger than 5 nm (the stopping limit of the technique).

The Reviewer is correct that a small peak at 39.5° 2θ corresponding to the Pt(111) facet was found in the PXRD patterns of Pt-Mo/NC_{0.02}. Accordingly, a small portion of metal nanoparticles up to about 4 nm in sizes was clearly observed in the HAADF-STEM images (**Fig. 2e**). Actually, this is within the detection limit of the PXRD technique, which is generally applicable to nanocrystalline powders with crystallite size above several nanometers (>3 nm, *Material Characterization*, 58 (2007) 883-891, *ChemPlusChem*, 88 (2023) e202300111; >2-2.5 nm, *Catal. Lett.*, 145 (2015) 777). Even for the crystallites below 2-3 nm, diffraction patterns might be still visible, but probably displaying more broad diffraction peaks. For instance, broad diffraction peaks of CdS sphere particles of 1 and 2 nm were reported by Holder and Schaak (*ACS Nano*, 13 (2019) 7359-7365). O'Connell and Regalbuto also found the diffraction lines of Au particle as small as 1.2 nm (*Catal. Lett.*, 145 (2015) 777-783). The Reviewer is kindly asked to the below figures cited from the above references.

Fig. 2 HRTEM images and particle sizes of (a) 3.0 wt% AuNp/VXC, (b) Au/KBB, (c) Au/VXC, and (d) high magnification image of Au/KBB (Figure and captions cited from Catal. Lett., 145 (2015) 777-783)

Fig. 4 XRD profile Au/VXC using D/teX Ultra detector (Figure and captions cited from Catal. Lett., 145 (2015) 777-783)

Figure 2. Simulated powder X-ray diffraction patterns for wurtzite CdS spherical particles of different sizes that range from 1 μm to 1 nm. The inset shows the 1, 2, and 5 nm XRD patterns on an expanded y-axis scale for clarity. (Figure and captions cited from ACS Nano, 13 (2019) 7359-7365)

5. However, the authors propose that the average size of the metallic particles, calculated by HR-TEM, figure 2e, is 0.66 nm. How do the authors explain this controversy?

The Reviewer is kindly reminded that the HAADF-STEM images showed the full size distributions of the metal species. The average particle size of 0.66 nm was the statistic number derived from the fitting line. In this specific catalyst, a small portion of larger nanoparticles up to about 4 nm were also observed which should be responsible for the diffraction of the Pt(111) facet.

6. 4. The authors claim that the post-reaction sample shows no apparent modifications. However, there is an accentuation of the peak (110) of amorphous carbon, which could indicate a partial graphitisation of the carbonaceous support. On the one hand, the authors should show the XRD diffractogram from 10-90° 2 Theta, to observe the first carbon diffraction peak, and be able to compare them before and after the reaction, correlating these modifications when calculating the La and Lc parameters, and the packing factor, with the possible presence of carbonaceous leftover species.

Thank you for your valuable comments. Since the metallic species are generally regarded as the active sites in the decomposition of HCOOH, we mainly focused on these in the previous manuscript. Following your suggestion, **we have replotted the PXRD patterns showing the range at 10-80° 2 θ (supplementary Fig. 16a)**. Besides, **we have estimated the La and Lc values for the carbon supports before and after the catalytic tests, and these parameters are not significantly different**. To further exclude the presence of carbonaceous deposits, **we have performed additional Raman spectra analyses, which showed negligible changes between the fresh and used catalysts (supplementary Fig. 16b)**.

Accordingly, we have been rephrased the sentences as follows:

Page 15, lines 18-22: 'Pt-Mo/NC_{0.07} after the stability test was thoroughly characterized by different techniques. PXRD and Raman spectra analysis revealed the same amorphous nature

of the spent catalyst and **no significant alternation of the carbon carrier (Supplementary Fig. 16). No diffractions of Pt- and/or Mo-related compounds were detected by PXRD (Fig. S11), suggesting that these metal species remained highly dispersed.**

The Scherrer formula is used to obtain the crystallite height (Lc) and the crystallite width (La) :

$$L_c = (\kappa \times \lambda) / (\beta_{002} \times \cos \theta_{002}),$$

$$L_a = (\kappa \times \lambda) / (\beta_{100} \times \cos \theta_{100}),$$

wherein κ is the Scherrer constant (assuming 0.89), λ is the wavelength (0.154056 nm), β is the full width at half maximum, and θ is the Bragg angle.

These values were calculated to be $L_c = 0.93$ nm, $L_a = 1.59$ nm for the fresh catalyst, and $L_c = 0.87$ nm, $L_a = 1.51$ nm for the use catalyst.

Fig. S11. ~~PXRD patterns and of Pt-Mo/NC_{0.07} after the 50 h stability test in formic acid decomposition.~~

Supplementary Fig. 16. a, PXRD patterns and, b, Raman spectra of Pt-Mo/NC_{0.07} after the 50 h stability test in formic acid decomposition.

For this reason, the work might not be published in Nature communications.

Taken all the valuable comments and suggestions of the three Reviewers, **we have provided additional experimental/theoretical evidences to support the main claims of this work, and fully revised the manuscripts.** We trust that these actions should greatly strengthen the novelty and clarity of this piece. We hope that the Reviewer 1 will now be satisfied with our actions in this revision and support the publication of this revised piece in Nature communications.

Reviewer #2

What are the noteworthy results?

The authors reported a defect-driven nanostructuring strategy through combining defect engineering of nitrogen-doped carbons and sequential metal depositions to prepare a series of Pt and Mo ensembles ranging from single atoms to sub-nanoclusters. When applied in continuous gas-phase decomposition of formic acid, the low-nuclearity ensembles with unique Pt₃Mo₁N₃ configuration deliver CO-free hydrogen at full conversion with high activity and good stability.

Will the work be of significance to the field and related fields? How does it compare to the established literature? If the work is not original, please provide relevant references. The reported work is interesting because it can produce CO-free hydrogen over 50 hours from formic acid. However, to have a significant impact on hydrogen applications using formic acid as the energy carrier, the authors need to consider the following:

The Reviewer 2 is warmly thanked for the very careful assessment of our work and for recognizing the merits of this piece. We also thank you for your critical comments that are extremely valuable for us to further improve the quality of our manuscript. In this revision, we have conducted additional experiments and DFT calculations to address your concerns and a detailed point-to-point reply has been provided as follows:

(1) Operating temperature: Even though the volumetric energy density of formic acid is higher than hydrogen gas, it is still very low compared to other liquid hydrogen carriers. The significant advantage of formic acid as a hydrogen carrier over other liquid hydrogen carriers (e.g., methanol or ethanol) is that it can reform at room temperature. As the operating temperature increases beyond the room temperature, the advantage of using formic acid as a hydrogen carrier diminishes. In order to have a significant impact, it is important that the catalyst needs to give both high activity and good stability at the ambient conditions (e.g., 25 °C and 1 atm) for the formic acid decomposition reaction. The reported Pt-Mo/NC catalyst is interesting, but it still requires a very high temperature of 100 °C to provide good performance. The real practical technological challenge for the formic acid decomposition reaction in hydrogen application is developing non-noble metal catalysts (or ultra-low precious metal catalysts) that can deliver CO-free hydrogen at ambient conditions over a long time.

We fully agree with the Reviewer's opinions. Indeed, the activity of the best-developed catalyst Pt-Mo/NC_{0.07} markedly deteriorated below 100 °C, and it is still far from the viewpoint of practical applications. Nonetheless, it still represents one of the best solid catalysts for the gas-phase dehydrogenation of HCOOH reported to date. Regarding the utilization efficiency of precious metals, Pt-Mo/NC_{0.07} outperformed the other gas-phase dehydrogenation catalysts by ca. one order of magnitude in the total metal-based reaction rates, and reached close or even better values than those of the state-of-the-art liquid-phase dehydrogenation catalysts. (The Reviewer is kindly asked to refer to **Fig. 3d** and **Supplementary Table 1**).

As pointed by the Reviewer, the development of non-noble metal catalysts is a more practical direction. In this context, we trust that the principle of the proposed strategy of defect-driven nanostructuring of bimetallic low-nuclearity ensembles might stimulate a pulse for the delicate design of advanced catalytic materials comprising multi-atoms and bring new opportunities to tackle this challenge. **These messages have been accommodated in the revised Conclusion section.**

Page 25, lines 15-19: 'To develop a practical HCOOH-to-H₂ technology, it is still imperative to design more efficient catalytic materials based on non-noble metals which can operate ideally at ambience conditions. In this scenario, the developed defect-driven nanostructuring approach may offer new opportunity to tackle this challenge through rational design of more sophisticated low-nuclearity heteroatom ensembles.'

(2) CO-free hydrogen: The gas stream from the reformer has to be free from CO gas (<10 ppm), or the catalytic performance of the fuel cells will be degraded significantly. Based on the given experimental data, it is hard to determine whether there were any ppm levels of CO gas or not. The author should provide information about CO gas concentration in ppm levels.

Thank you for your valuable comments. In our previous analysis method, N₂ was used as the carrier gas for the gas chromatograph (GC). As can be seen from supplementary **Fig. 15a**, the trace CO can hardly be detected. However, the zoom figure in supplementary **Fig. 15a'** showed a small bump around 5.5 min, which should be contributed by CO. Nonetheless, this very small peak was intervened by the tailing of N₂ signal and thus the area could not be well integrated. To overcome this drawback, we have replace N₂ with He as the carrier gas for the GC analysis. Also, the full spectra show no obvious CO signals (**Supplementary Fig. 15b**), while tiny CO peaks can indeed be found in the zoomed figure (**Supplementary Fig. 15b'**). By referring to the standard gases, **we have now quantified the trace CO during stability test on Pt-Mo/NC_{0.07}. The average CO content was determined to be 15.5 ppm.** Accordingly, **we added the quantification method of trace CO in the Experimental section, and rephased our claim by replacing the term 'CO-free hydrogen' with 'high-purity hydrogen'.**

Page 3, lines 7-11: *When applied in continuous gas-phase decomposition of formic acid, the low-nuclearity ensembles with unique Pt₃Mo₁N₃ configuration deliver high-purity CO-free hydrogen at full conversion with unexpected high activity of 0.62 mol_{HCOOH} mol_{Pt}⁻¹ s⁻¹ and remarkable stability, significantly outperforming the previously reported catalysts.*

Page 15, lines 15-18: 'The activity gradually increased after a few hours of stabilization and slightly fluctuated at ca. 90-96%, while trace CO of 15.5 ppm in average CO₂ was detected the only detectable product in our gas chromatography (**Supplementary Fig. S15**).'

Page 31, lines 7-8: 'Trace CO was quantified by referring the standard gases (10 and 50 ppm CO/He).'

Supplementary Fig. 15. The gas chromatograph (GC) profiles of the products in HCOOH decomposition on Pt-Mo/NC_{0.07} during stability test, together with the profiles of standard gases (10 and 50 ppm CO in He). Carrier gases used in the GC analysis: **a**, N₂, **b**, He. **a'** and **b'** are the zoomed profiles in **a** and **b**, respectively.

Does the work support the conclusions and claims, or is additional evidence needed? (1) The authors stated that “Mo atoms are first strongly coordinated by the N defects followed by the deposition of Pt, wherein controlled aggregation of Pt is realized by simply reducing the numbers of nitrogen defects.” The authors need to clearly articulate how the N defects can coordinate with Mo atoms, and how this coordination chemistry can be affected by the concentration of N doping.

Thank for you pointing out the questions that have not been well considered previously. To understand how the N defects can coordinate with the Mo atoms, **we resolve to DFT to compare the formation energies of Mo species stabilized by different N defects (Supplementary Fig. 10)**. As our characterization showed the more prominent role of pyridinic N in anchoring the metal atoms, multiple pyridinic-N defective sites were considered that may also reflects the increasing concentrations of N doping. As shown in **Supplementary Fig. 10a**, the

formation energy of the MoN_x entities increased with the increasing numbers of N anchors, reflecting the increasing stability. Bader charge analysis reveals the electron-deficient states of the Mo atoms in all these configurations (**Supplementary Fig. 10b**), agreeing well with the ionic nature of Mo based on the Mo 3d XPS and XANES observations.

Supplementary Fig. 10. a, The formation energy and Bader charge of different MoN_x ($x = 2-4$) entities, and, **b**, the configurations and charge density plots of MoN_x .

To further address the impact of the numbers of N defect on the coordination chemistry of the Mo species, we have prepared additional two Mo/NC_x catalysts (5 wt.% Mo, $x = 0.004$ and 0.02) as references following the same recipe for $\text{Mo/NC}_{0.07}$. HAADF-STEM images clearly show that a few particles were formed on $\text{Mo/NC}_{0.02}$ and more severe aggregation occurs for $\text{Mo/NC}_{0.004}$ (Supplementary Fig. 11). The lattice fringes corresponding to the (101) facets of Mo_2C is verified for $\text{Mo/NC}_{0.004}$. These observations were further corroborated by PXRD and Mo 3d XPS analyses (Supplementary Fig. 12). Specifically, several diffraction facets corresponding to Mo_2C were observed on $\text{Mo/NC}_{0.004}$, and the contribution of surface $\text{Mo}^{2+}/\text{Mo}^{4+}$ species were also confirmed for this sample. Altogether, the above findings suggest that the N defect plays a critical role in stabilizing the Mo species against sintering.

Supplementary Fig. 11. The HAADF-STEM images of $\text{Mo/NC}_{0.02}$ and $\text{Mo/NC}_{0.004}$.

Supplementary Fig. 12. a, PXRD and, **b**, Mo 3d XPS spectra of Mo/NC_y and commercial Mo₂C. The vertical lines in **a** show the reference standard of Mo₂C.

The above new results have been accommodated into the revised manuscript.

Page 13, lines 6-21: ‘To understand how the N defects can coordinate with the Mo atoms, the formation energies (E_f) as well as Bader charges of Mo species stabilized by multiple pyridinic-N defects were calculated by DFT (Supplementary Fig. 10). A higher E_f was observed with the increasing numbers of the N anchor in the MoN_x entities, suggesting the higher stability. Bader charge analysis revealed the electron-deficient states of the Mo atoms in all these configurations, agreeing well with the Mo 3d XPS and XANES observations for the Mo/NC_{0.07} and Pt-Mo/NC_{0.07}. To further address the impact of the numbers of N defect on the coordination chemistry of the Mo species, two additional Mo/NC_x catalysts (5 wt.% Mo, $x = 0.004$ and 0.02) were prepared as references following the same recipe for Mo/NC_{0.07}. HAADF-STEM images clearly showed that a few particles were formed on Mo/NC_{0.02} and more severe aggregation occurred for Mo/NC_{0.004} (Supplementary Fig. 11). The lattice fringes corresponding to the (101) facets of Mo₂C were verified for Mo/NC_{0.004}. These observations were further corroborated by PXRD and Mo 3d XPS analyses (Supplementary Fig. 12). Specifically, several diffraction facets corresponding to Mo₂C were observed on Mo/NC_{0.004}, and the contribution of surface Mo²⁺/Mo⁴⁺ species were also confirmed for this sample. Altogether, the above findings confirmed the critical role of planar N defect in stabilizing the Mo species against sintering.’

Furthermore, there are no clear experimental data and analysis to explain how these N defect-controlled Mo atoms interact with Pt atoms and control the final ensemble size of Pt. What is the exact role of Mo atoms in controlling the ensemble size of Pt in the overall synthesis process?

Thank you for your good question. We think that the key role of N-coordinated Mo species in influencing the ensemble sizes of Pt species is through modulating the number of N defects available for coordination with Pt atoms. This is supported by the following two experimental observations:

- (1) More severe aggregation of Pt species is observed for the bimetallic Pt-Mo/NC_{0.07} catalysts as compared with Pt/NC_{0.07},
- (2) and the sizes of Pt-Mo ensembles grow at decreasing numbers of the N defects in the order of Pt-Mo/NC_{0.13} < Pt-Mo/NC_{0.07} < Pt-Mo/NC_{0.02}.

Since our characterizations suggest the atomic metal dispersion on both Mo/NC_{0.07} and Pt/NC_{0.07}, and the clustering is only observed for Pt-Mo/NC_{0.07}, we trust it is reasonable to attribute the growing Pt sizes of Pt-Mo/NC_{0.07} to the reduced number of the free N defects for coordination. Following this reasoning, it can also well explain the increasing Pt sizes at the decreasing N defects for the bimetallic catalyst series. As we adopted the sequential deposition method to prepare the bimetallic catalysts (first Mo and then Pt), once the N defects are coordinated with the Mo atoms, there are no sufficient N sites for the coordination of Pt atoms that are consequently more prone to aggregation.

On the other hand, we also agree with the Reviewer that the N-coordinated Mo atoms might interact with the Pt species and potentially influence the clustering behavior. **We have tentatively compared the formation energy of Pt₄N₃ and Pt₃Mo₁N₃ with similar tetrahedron configurations (Supplementary Fig. 14).** The results show that Pt₃Mo₁N₃ possesses a slightly higher formation energy than Pt₄N₃ (-16.84 vs. -16.34 eV), suggesting that the presence of Mo is beneficial for the formation of low-nuclearity Pt-Mo ensembles.

Supplementary Fig. 14. The configurations of Pt₄N₃ and Pt₃Mo₁N₃, accompanied with the formation energy.

Taking into all the above discussions, **we stick to our previous hypothesis on the key role of N-coordinated Mo atoms, but also stressed the potential role of Pt-Mo interactions in affecting the clustering behavior for which more evidences are needed.**

We have accommodated the new calculated results in the revised piece.

Page 14, lines 7-11: *'In addition, we have tentatively compared the E_f of Pt₄N₃ and Pt₃Mo₁N₃ with similar tetrahedron configurations (Supplementary Fig. 14). The results showed that Pt₃Mo₁N₃ possesses a slightly higher E_f than Pt₄N₃ (-16.84 vs. -16.34 eV), suggesting that the presence of Mo is thermodynamically beneficial for the formation of low-nuclearity Pt-Mo ensembles.'*

(2) Based on the performance data, Pt-Mo/NC_{0.07} shows an improved activity compared to Pt/NC_{0.07}. This improved activity could be originated from the ensemble size effect and/or the presence of Mo atoms. As the ensemble size changes and Mo atoms are introduced to Pt clusters, both their physicochemical and electronic properties will change and affect the catalytic performances. The authors provided DFT calculation data to explain the ensemble size effect and role of Mo atoms. However, their DFT data do not sufficiently address the size effect and the nature of the synergistic catalysis of the Pt-Mo ensembles. Could they prepare the samples with and without Mo while fixing the ensemble size? This comparison would allow them to isolate the Mo effect for example.

Thank you for your comments and valuable suggestions. To address these questions, two actions have been taken in this revision:

First, **we have attempted to prepare Pt/NC catalysts with the average Pt particle sizes close to those of Pt-Mo/NC_{0.07}, following the same recipe.** Since Pt/NC_{0.07} in the absence of Mo showed predominantly single Pt atoms even after reduction at a high temperature of 973 K, we have adopted NC_{0.02} as the carrier because we envision that the lower number of N defects may facilitate the agglomeration of Pt species. By carefully tuning the reduction temperature of 573 K, We obtained a sample - Pt/NC_{0.02} with similar particle size distributions as those of Pt-Mo/NC_{0.07}, with an average size of 0.31 ± 0.27 nm (Supplementary Fig. 26).

Supplementary Fig. 26. HAADF-STEM images of Pt/NC_{0.02} and the particle size distributions. This reference catalyst was prepared following the same recipe as Pt/NC_{0.07} but with a lower

reduction temperature of 573 K in order to reach similar particle size distributions as those of Pt-Mo/NC_{0.07}.

Then, we evaluated the catalytic performance of Pt/NC_{0.02} in HCOOH decomposition under the same reaction conditions (Supplementary Fig. S27). The results showed that Pt-Mo/NC_{0.07} still displayed higher activity than Pt/NC_{0.02}. More importantly, the CO₂ selectivity was significantly higher on Pt-Mo/NC_{0.07}. These observations thus further support the Pt-Mo synergy in enhancing the HCOOH dehydrogenation.

Supplementary Fig. 27. Comparison on the catalytic performance between Pt/NC_{0.02} and Pt-Mo/NC_{0.07} in HCOOH decomposition.

Second, we have constructed a model of Pt₄N₃ cluster with a similar configuration as Pt₃Mo₁N₃, computed the adsorption and dissociation energies of HCOOH (Supplementary Fig. 21), and simulated the reaction coordinate on the Pt cluster by DFT (Supplementary Fig. 25). Comparison on the adsorption and dissociation energies showed that Pt₃Mo₁N₃ is still more favorable than Pt₄N₃ for the HCOOH activation. Furthermore, the reaction simulation showed comparable energy barriers for the dissociation of the H atom in *HCOO and the desorption of *COO between Pt₄N₃ and Pt₃Mo₁N₃, however, the activation of HCOOH is more energy-demanding on the former (0.52 vs. -0.22 eV). These might explain the higher activity of Pt-Mo/NC_{0.07} than the Mo-free Pt catalysts with comparable particle sizes, and thus supporting the presence of Pt-Mo synergy.

Fig. S17. The **a**, adsorption and **b**, dissociation energies of HCOOH at different Pt model systems. The corresponding side-view configurations were shown in the insets.

Supplementary Fig. 21. The adsorption and dissociation energies of HCOOH at different Pt model systems, accompanied with the corresponding side-view configurations.

Supplementary Fig. 25. **a**, The relative energy of HCOOH decomposition on the Pt_4N_3 and $\text{Pt}_3\text{Mo}_1\text{N}_3$, and **b**, the side view of DFT-optimized adsorption configurations of the intermediates. The energy barriers in **a** were highlighted by the bold numbers.

These new findings have been accommodated in the revised piece.

Page 24, lines 2-11: ‘To further support the synergistic effect, DFT simulations on the other monometallic sites, i.e., MoN_3 (single Mo atom coordinated with triple pyridinic-N sites) and Pt_4N_3 (tetrahedron Pt clusters stabilized in triple pyridinic-N sites) were performed. Comparison on the reaction coordinates revealed higher energy barriers for i) the activation of $^*\text{HCOO}$ and desorption of $^*\text{COO}$ on MoN_3 (Supplementary Fig. 24), and ii) the deprotonation of HCOOH on Pt_4N_3 , as compared with the respective elemental steps on the $\text{Pt}_3\text{Mo}_1\text{N}_3$ sites (Supplementary Fig. 25). To further substantiate the theoretical findings, a Mo-free reference – $\text{Pt}/\text{NC}_{0.02}$ with similar particle size distributions (0.31 ± 0.27 nm, Supplementary Fig. 26) as those of $\text{Pt-Mo}/\text{NC}_{0.07}$ was prepared and evaluated in HCOOH decomposition, which indeed displayed lower activity and much poorer CO_2 selectivity (Supplementary Fig. 27).’

(3) The authors stated that “DB-FTIR study evidenced the different kinetic fingerprints of formic acid molecules with the distinct mono- and bimetallic sites as illustrated in Fig. S16.” However, this reviewer cannot see how the DB-FTIR data shown in Figure 4 can lead to Figure S16. The DB-FTIR data do not reveal the distinct mono- and bimetallic sites. Instead, they simply show the different fingerprints of formic acid and its derivatives as the function of temperature. Based on the DB-FTIR data, it is very hard to make any conclusive statements about the degree of

interaction between the specific sites of the catalyst and formic acid (and its derivatives). For example, the authors stated that “On the representative single-atom Pt sites, strong adsorption of formic acid was observed, but the deprotonation was much difficult.” How can the authors prove that the deprotonation (of first H from O-H or second H from C-H?) was harder over the single-atom Pt sites than other catalysts?

Thank you for the question. The intensity of the absorption bands in DB-FTIR spectra can reflect the strength of the interactions of the surface adsorbed species with the catalyst surface. According to the previous literatures, the absorption peaks at 1718 and 1595 cm^{-1} can be assigned to the C–O vibration of the molecularly adsorbed HCOOH_{ad} and the O–C–O vibration due to the adsorbed formate species (HCOO_{ad}), respectively. With the increasing of reaction temperatures, the bands related to HCOOH_{ad} gradually decreased in intensity for all the three catalysts. However, the bands associated with HCOO_{ad} remained essentially stable on single-atom $\text{Pt}/\text{NC}_{0.07}$, but gradually attenuated on $\text{Mo}/\text{NC}_{0.07}$ and $\text{Pt-Mo}/\text{NC}_{0.07}$. These observations at least suggest that the protonation of the second H atom in HCOO_{ad} was much difficult for $\text{Pt}/\text{NC}_{0.07}$. To avoid any misunderstanding, **we have revised the claim** “On the representative single-atom Pt sites, strong adsorption of formic acid was observed, but the deprotonation **of the second H atom in HCOO_{ad} was much difficult**” (Page 20, Lines 7-9). On the other hand, we agree with the Reviewer that the DB-FTIR results cannot sufficiently lead to the conclusions drawn in Fig. S16. To avoid over-interpretating the messages, **we have removed this figure in the revised piece**. ‘In general, the above DB-FTIR study evidenced the different kinetic fingerprints of formic acid molecules with the distinct mono- and bimetallic sites **as illustrated in Fig. S16.**’ (Page 20, Lines 5-7)

Fig. S16. Schematic illustrations for the adsorption and dissociation of formic acid on the representative catalysts based on the DB-FTIR results. Top: $\text{Pt}/\text{NC}_{0.07}$, middle: $\text{Pt-Mo}/\text{NC}_{0.07}$, bottom: $\text{Mo}/\text{NC}_{0.07}$

(4) The authors assigned the adsorption bands of the DB-FTIR data at 1718 and 1595 cm^{-1} as the C-O vibration of the molecularly adsorbed HCOOH and the O-C-O vibration due to the adsorbed formate species. If this is true, as the temperature increases the peak intensity of HCOOH_{ad} should decrease while the peak intensity of HCOO_{ad} should increase followed by the decrease. Do the DB-FTIR data show these trends? In general, the interpretation of the DB-FTIR data needs to be significantly improved and better match them to the predicted DFT-derived reaction mechanisms.

Thank you very much for these comments. The assignments of the absorption peaks at 1718 and 1595 cm^{-1} to the C-O vibration of the molecularly adsorbed HCOOH (HCOOH_{ad}) and the O-C-O vibration due to the adsorbed formate species (HCOO_{ad}) were based on the previous literature results (Ugrai, *et al.*, *J. Catal.*, 279 (2011) 213-219; Li *et al.*, *Chin. J. Catal.*, 29 (2008)105-107). In our DB-FTIR experiments (**Fig. R2**), the catalysts were first saturated with HCOOH vapor at room temperature before collecting the spectra. Once saturated, both physically and chemically adsorbed HCOOH species in significant amounts were readily detected on Pt-Mo/NC_{0.07}. The gaseous or physically adsorbed HCOOH_{ad} species can be gradually removed by purging with flowing N₂, while the chemically adsorbed HCOO_{ad} species did not markedly change in intensity (A slightly enhanced intensity of the peak at 1595 cm^{-1} was indeed found at the beginning during N₂ purging). We think the absence of obvious increasing intensity of the band at 1595 cm^{-1} might be due to relatively fast kinetics for the H dissociation. To slow down the kinetics, collecting the spectra at liquid N₂ temperature is an option. Nonetheless, the high freezing point of HCOOH (281 K) makes it unsuitable for such experiments.

To further substantiate the DB-FTIR results, **we have performed additional DFT simulations on the reaction mechanism of HCOOH dehydrogenation on single-site MoN₃ catalyst (Mo coordinated with triple pyridinic-N defects)**. In general, the DB-FTIR results fit well with the DFT calculations. The Reviewer is kindly asked to refer to the replies to Question 5.

Fig. R2. The DB-FTIR spectra of HCOOH adsorption on Pt-Mo/NC_{0.07} after different treatments.

(5) Based on operando DB-FTIR data (Figure 4 c and d), it seems that both Mo/NC_{0.07} and Pt-Mo/NC_{0.07} show very similar spectra. However, their activities are very different (Pt-Mo/NC_{0.07} shows the best performance while Mo/NC_{0.07} shows the worst performance). To explain this, the

authors speculated that the abstraction of another H atom might be more energy-demanding than the dissociation of HCOOH. To support its speculation, the authors should show the energy profiles for the formic acid decomposition over the Mo clusters or Mo single atoms (whichever best represents the Mo/NC0.07 sample).

Thank you for your comments. To support our claim, **we have simulated the reaction coordinate of HCOOH decomposition on MoN₃ sites (Mo coordinated with triple pyridinic-N defects) by DFT (Supplementary Fig. 24)**. We found that the activation of the first H atom in HCOOH is relatively easier with a low energy barrier of 0.19 eV, while the energy barriers for the deprotonation of the second H atom in HCOO* and the desorption of COO* species are 1.59 and 1.35 eV, respectively. These findings thus fully support our speculations from DB-FTIR, that “*The mono-metallic Mo catalyst also showed a higher propensity toward HCOOH adsorption and its dissociation but displayed the poorest decomposition activity. This suggested that other fundamental steps, such as the abstraction of another H atom, might be more energy-demanding than the dissociation of HCOOH (vide infra)*”. Another explanation could be the too strong adsorption of HCOOH on the Mo⁶⁺ sites as indicated by the much higher desorption temperatures in the operando DB-FTIR study.” **(Page 20, Lines 9-14)**

Furthermore, comparison of the energy profiles shows that both the activation of the H atom in the adsorbed HCOO* species and the desorption of the surface COO* species were more energy-demanding on MoN₃ than on Pt₃Mo₁N₃ (1.59 vs. 1.18 eV, and 1.35 vs. 1.17 eV). These findings also support our claim, that “*In contrast, these two fundamental steps of formic acid adsorption and dissociation were both more favorable on the low-nuclearity Pt-Mo ensembles, thus accounting for the highest activity*” **(Page 20, Lines 14-16)**. **We have strengthened the above claims by accommodating the new DFT calculation results in the revision.**

Supplementary Fig. 24. **a**, The relative energy of HCOOH decomposition on the MoN_3 single-atom model, and **b**, the side view of DFT-optimized adsorption configurations of the intermediates. The reaction simulation on $\text{Pt}_3\text{Mo}_1\text{N}_3$ was provided in **a** for reference.

(6) For the DFT energy profile of $\text{Pt}_3\text{Mo}_1\text{N}_3$, the second H (from C-H bond) is deprotonated by chemisorbing to the nearest Pt site. However, in order for this H transfer to occur, this H must first move closer to the nearest Pt site. Because this H (from C-H) is located too far away from the nearest Pt site, this reviewer is not entirely convinced that such a transfer reaction can occur with the energy downhill. This should be a very unfavorable reaction because adsorbed HCOO needs to be stretched out to the nearest Pt site in an extreme degree.

The Reviewer is fully correct. As can be seen from **Fig. 5**, the deprotonation of the second H on $\text{Pt}_3\text{Mo}_1\text{N}_3$ occurs in multi-steps ($\text{nt1} \rightarrow \text{S4} \rightarrow \text{TS}$), and is highly energy-demanding (1.18 eV). The transfer of the H atom to the nearby Pt site first involves the orientation of the adsorbed $^*\text{HCOO}$ species with the H pointing toward the Pt site (see $\text{Int1} \rightarrow \text{S4}$, **Supplementary Fig. 22c**) as predicted by the Reviewer. For the conciseness purpose, **Fig. 5** only presents the adsorption configurations for the key steps, while the full elementary steps are shown in the **Supplementary Fig. 22c**,

Fig. 5. Mechanistic insights from DFT. **a**, Energy profiles for the decomposition of formic acid on the different model catalysts. **b**, Side view of DFT-optimized key adsorption configurations.

Supplementary Fig. S1822. Side view of DFT-optimized adsorption configurations of the intermediates on **a**, Pt(111), **b**, Pt₁N₃, and **c**, Pt₃Mo₁N₃, for the reaction profiles in in Fig. 5.

Reviewer #3

The authors reported defect-driven nanostructuring strategy for the preparation of well-dispersed bimetallic Pt-Mo species on nitrogen-doped carbon, which have shown good catalytic activity in gas-phase formic acid dehydrogenation.

Although the investigation comprises a systematic study of the material properties, catalytic evaluation, and mechanism, the following issues should be addressed before consideration for publication.

We highly appreciate the Reviewer 3 for the careful evaluation of this work, and thank you for your support of publication of this piece after addressing the following issues. The detailed replies to your questions have been provided point-by-point as follows:

1. From the XPS spectra, the authors suggested the presence of Pt-N interactions. However, CH₄ was used during the reduction process, which could lead to the formation of metal carbides (Catal. Sci. Technol., 2020, 10, 6790–6799). It is thus important to show whether Pt-C or Mo₂C was formed.

Thank you for the relevant comment. As pointed out by the Reviewer, Mo₂C is a potential product when Mo-based materials are carbonized in CH₄-containing stream at high temperatures. According to the literatures (*Inorg. Chem.*, 62 (2023) 653-658, *ACS Sustain Chem. Eng.*, 7 (2019) 18375-18383, *Dalton Trans.*, 51 (2022) 17547-17552), the presence of Mo₂C species can be detected by using XPS. Typical Mo²⁺ species at the binding energy of ca. 228.3-228.7 eV, corresponding to Mo 3d orbital, have been reported. This has also been confirmed from the XPS result of a commercial Mo₂C sample (**Supplementary Fig. 12b**). On the contrary, the Mo 3d XPS spectra of all our Mo-containing samples showed the exclusive Mo⁶⁺ species at a much higher binding energy of 232.4-232.8 eV. Therefore, we trust that the Mo atoms were mainly coordinated with the N defects. To further understand the potential role of N defects in stabilizing the Mo atoms, **we have intentionally tune the numbers of N defect in NC_x carriers, which were then used to prepared additional Mo/NC_x catalysts following the same recipe of Mo/NC_{0.07}**. Our thorough characterization results combining HAADF-STEM, PXRD, AND XPS showed that Mo₂C can be formed only for carrier with the least number of N defect, *i.e.*, NC_{0.004} (**Supplementary Figs. 11,12**). These observations highlight the importance of sufficient N defective sites in stabilizing the Mo atoms and against sintering during the high-temperature annealing treatment. In addition, **we have performed additional DFT calculations to understand how the numbers of N defects can influence the stabilization of single Mo atoms (Supplementary Fig. 10)**. As our characterization showed the more prominent role of pyridinic N in anchoring the metal atoms, multiple pyridinic N defective sites were considered that may also reflect the increasing concentrations of N doping. As shown in **Supplementary Fig. 10a**, the formation energy increased with the increasing numbers of N atoms in the MoN_x entities, reflecting the increasing stability. Bader charge analysis reveals the electron-deficient states of the Mo atoms in all these configurations, agreeing well with the ionic nature of the Mo species based on Mo 3d XPS and XANES observations.

The above new results have been accommodated into the revised manuscript.

Page 13, lines 6-21: 'To understand how the N defects can coordinate with the Mo atoms, the formation energies (E_f) as well as Bader charges of Mo species stabilized by multiple pyridinic-N defects were calculated by DFT (Supplementary Fig. 10). A higher E_f was observed with the increasing numbers of the N anchor in the MoN_x entities, suggesting the higher stability. Bader

charge analysis revealed the electron-deficient states of the Mo atoms in all these configurations, agreeing well with the Mo 3d XPS and XANES observations for the Mo/NC_{0.07} and Pt-Mo/NC_{0.07}. To further address the impact of the numbers of N defect on the coordination chemistry of the Mo species, two additional Mo/NC_x catalysts (5 wt.% Mo, x = 0.004 and 0.02) were prepared as references following the same recipe for Mo/NC_{0.07}. HAADF-STEM images clearly showed that a few particles were formed on Mo/NC_{0.02} and more severe aggregation occurred for Mo/NC_{0.004} (Supplementary Fig. 11). The lattice fringes corresponding to the (101) facets of Mo₂C were verified for Mo/NC_{0.004}. These observations were further corroborated by PXRD and Mo 3d XPS analyses (Supplementary Fig. 12). Specifically, several diffraction facets corresponding to Mo₂C were observed on Mo/NC_{0.004}, and the contribution of surface Mo²⁺/Mo⁴⁺ species were also confirmed for this sample. Altogether, the above findings confirmed the critical role of planar N defect in stabilizing the Mo species against sintering.'

Supplementary Fig. 11. The HAADF-STEM images of Mo/NC_{0.02} and Mo/NC_{0.004}.

Supplementary Fig. 12. a, PXRD and, d, Mo 3d XPS spectra of Mo/NC_y and commercial Mo₂C.
 The vertical lines in a show the reference standard of Mo₂C.

Supplementary Fig. 10. a, The formation energy and Bader charge of different MoN_x (x = 2-4) entities, and, **b**, the configurations and charge density plots of MoN_x.

2. From the TEM studies, it was evident that some of the impregnated metal atoms were present as metal nanoparticles or metal clusters. However, surface sensitive techniques such as XPS did not show any Pt-Pt bond or Mo-Mo in the case of isolated impregnations (Pt/NC0.07 and Mo/NC0.07). These observations contradict the TEM studies, so the authors should discuss this anomaly.

Indeed, the Pt 4f and Mo 3d XPS spectra only showed the presence of Pt²⁺ and Mo⁶⁺ species on the surface of all the catalysts, and no metallic species can be observed although the presence of some Pt sub-nanoclusters was confirmed. This phenomenon can be explained by the quantum size effects, which are most prominent for small nanoparticles. As have been observed in many systems such as Pt (Schierbaum *et al. Surf. Sci.*, 345 (1996) 261) and CuO (Borghain, *Phys. Rev., B*, 61 (2000) 11093), the binding energies of the Pt 4f and Cu 2p spectra gradually increased when the sizes of the nanoparticles were reduced. The shift might be attributed to final state effects due to the reduced screening of the photo hole in very small particles (Lambert *et al., Catal. Lett.*, 82 (2002) 169). In our cases, these effect should be more prominent since the metallic ensemble sizes were very small (average sizes <0.66±0.76 nm). **A brief explanation has been added in the revision to clarify this phenomenon.**

Page 12, lines 9-11: 'The absence of surface Pt⁰ in Pt-Mo/NC_{0.02} with evidenced Pt clusters might be explained by the quantum size effect which is most prominent for small nanoparticles^{57,58}.'

3. Likewise, if the isolated impregnations do not show cluster formation, how can the bimetallic systems give rise to Pt-Pt clusters?

Thank you for your question. The Reviewer is kindly reminded that various NC carriers with different N:C ratios were adopted to disperse the active metal species. Pt/NC_{0.07} with a higher concentration of N defect showed predominantly single Pt atoms, while Pt clusters were increasingly formed for the bimetallic catalysts, Pt-Mo/NC_{0.07} and Pt-Mo/NC_{0.02}. The formation of Pt-Pt clusters in Pt-Mo/NC_{0.02} can be understood by the decreased N defects to stabilize the single Pt atoms. On the other hand, since we adopted sequential impregnation method to introduce first Mo and then Pt into the carriers for the bimetallic catalysts, the clustering of Pt in Pt-Mo/NC_{0.07} in comparison to Pt/NC_{0.07} was explained by that: the introduction of Mo species already possessed some N defectives sites, thus reducing the total numbers of these defective sites available for the stabilization of Pt in the next impregnation step. Actually, this is the key concept proposed in our work to tune the size of the bimetallic ensembles through combining defect engineering of the NC carriers with sequential impregnation method.

4. The cluster sizes of the Pt-Mo/NC_{0.07} (1-4 atoms) were estimated 10-30 times smaller than those of Pt-Mo/NC_{0.02} (13-133 atoms), but why did the TEM images of both materials seem to show a high degree of agglomerated cluster?

Indeed, clustering of Pt occurred for both Pt-Mo/NC_{0.07} and Pt-Mo/NC_{0.02} as noticed by the Reviewer. However, the degrees of agglomerated cluster showed marked differences. As can be seen from the HAADF-STEM images in **Fig. 2d,e**, the density of atomically dispersed metal species (the bright dots) was significantly lower on Pt-Mo/NC_{0.02} than on Pt-Mo/NC_{0.07}. The high magnification HAADF-STEM images (Supplementary **Fig. 6**) more clearly showed the severer agglomeration of metal species on Pt-Mo/NC_{0.02} than Pt-Mo/NC_{0.07}. This suggests the much higher clustering degree of the Pt-Mo/NC_{0.02} sample, and thus in line with the much bigger average particle sizes.

We need to mention that the images in Supplementary **Fig. 6** were mistakenly used, as they are the same as those displayed in **Fig. 2a-e**. **This mistake has now been corrected and new figures are provided in Supplementary Fig. 6 in the revised piece.**

Fig. S6. Additional HAADF-STEM images of the representative mono- and bi-metallic catalysts.

Supplementary Fig. 6. Additional HAADF-STEM images of the representative mono- and bi-metallic catalysts.

5. The FTIR studies suggested that Mo/NC may be a major active species (or partially carbonized MoxCy species) since both the Mo/NC0.07 and Pt-Mo/NC0.07 showed a similar dehydration pathway associated at elevated temperatures. This was contradicting with the plot “Fig. 3a” which shows 100% CO2 selectivity for Mo/NC0.07, while its FTIR shows minor quantity of CO-related peaks at 2190 cm⁻¹.

Thank you for your very delicate observation and your comments. Indeed, very subtle peaks at 2190 cm⁻¹ can be observed on Mo/NC_{0.07} at increasing temperatures. The intensity of these peaks was much weaker as compared with that observed on Pt-Mo/NC_{0.07}. The Reviewer is reminded that the CO selectivity was less than 0.5% at 473 K on Pt-Mo/NC_{0.07}. Given the much lower activity of Mo/NC_{0.07} as well as the extremely weak peak intensities at 2190 cm⁻¹, it is reasonable that trace CO cannot be detected by our GC (detection limit of 10 ppm). On the other hand, it might be also possible that the desorption of this specie adsorbed at 2190 cm⁻¹ is much difficult on Mo/NC_{0.07} than on Pt-Mo/NC_{0.07}. In this revision, **we have simulated the reaction coordinate of HCOOH dehydrogenation on MoN₃ sites (Mo coordinated with triple pyridinic-N defects) by DFT (Supplementary Fig. 24)**. Comparison of the energy profiles shows that both the activation of the H atom in the adsorbed *HCOO species and the desorption of the surface *COO species are more energy-demanding on MoN₃ than on Pt₃Mo₁N₃ (1.59 vs. 1.18 eV, and 1.35 vs. 1.17 eV). This might also partly explain why no CO was formed on Mo/NC_{0.07}.

Supplementary Fig. 24. **a**, The relative energy of HCOOH decomposition on the MoN₃ single-atom model, and **b**, the side view of DFT-optimized adsorption configurations of the intermediates. The reaction simulation on Pt₃Mo₁N₃ was provided in **a** for reference.

These new results have been added and discussed in the revision to support our claim.

Page 24, Lines 2-8: *To further support the synergistic effect, DFT simulations on the other monometallic sites, i.e., MoN₃ (single Mo atom coordinated with triple pyridinic-N sites) and Pt₄N₃ (tetrahedron Pt clusters stabilized in triple pyridinic-N sites) were performed. Comparison on the reaction coordinates revealed higher energy barriers for i) the activation of *HCOO and desorption of *COO on MoN₃ (Supplementary Fig. 24), and ii) the deprotonation of HCOOH on Pt₄N₃ (Supplementary Fig. 25), as compared with the respective elemental steps on the Pt₃Mo₁N₃ sites.'*

6. The authors stated that "gas-phase dehydrogenation is attractive considering the mild exothermicity and ease of catalyst separation as compared with the endothermic liquid-phase dehydrogenation (-15 vs. 29 kJ mol⁻¹"); however, vapor-phase dehydrogenation is operated at relatively higher temperatures. In the case of dilute feed (10% FA in water), it also generated unwanted steam, which is an endothermic process. Since 90% of water is being used in the feed, a significant amount of energy will be lost for vaporization. Have the authors proposed any heat recovery mechanism? If an inert gas such as helium is used as a carrier gas, are additional separation steps needed for hydrogen purification?

Thank you for your questions. Although diluted aqueous solutions of HCOOH, sometime in conjunction with additives such as sodium formate, are frequently used in the liquid-phase decomposition, we fully agree with the Reviewer 2 on that it is an ideal approach to use pure HCOOH for the gas-phase dehydrogenation. As pointed out by the Reviewer, this is a more energy-saving way to avoid additional energy input for water heating or separation. For our experiments, HCOOH/H₂O solution was adopted simply because we were not able to deliver an extremely low feed of pure HCOOH in a stable manner with the syringe pump available in our lab. As an alternative, a diluted HCOOH solution was adopted instead in order to ensure a more reliable feeding rate of HCOOH.

Minor comments:

7. The authors need to clarify whether metal loading is in weight percentage or atomic percentage. Text and table values were given in different units.

The metal loadings reported were in weight percentage based on ICP analysis. We are sorry for the inconsistency. **The unit of the metals in Supplementary Table 2 has been changed from at.% to wt.% in the revision.**

Supplementary Table 2. Characterization data of the NC hosts and the corresponding metal-supported catalysts.

Sample	V_{total}^a / (cm ³ g ⁻¹)	S_{BET}^b / (m ² g ⁻¹)	C ^c / at. %	N ^c / at. %	Pt ^d / at. % wt. %	Mo ^d / at. % wt. %	N:C ^e / mol mol ⁻¹
NC _{0.13}	0.32	491	62.17	9.42	-	-	0.13(0.15) ^c
NC _{0.07}	0.26	464	74.86	5.93	-	-	0.07(0.08) ^c
NC _{0.02}	0.38	571	88.42	2.06	-	-	0.02(0.02) ^c
Pt/NC _{0.07}	0.90	182	-	-	0.54	-	-
Mo/NC _{0.07}	0.40	347	-	-	-	5.56	-
Pt-Mo/NC _{0.07}	0.72	227	-	-	0.48	5.04	-
Pt-Mo/NC _{0.13}	0.73	190	-	-	0.46	4.87	-
Pt-Mo/NC _{0.02}	0.34	288	-	-	0.43	4.84	-

^aDetermined from the amount of N₂ adsorbed at $p/p_0 = 0.97$. ^bBET method. ^cXPS. ^dICP-OES.

^eC,H,N elemental analysis.

8. The references about the reviews for formic acid dehydrogenation should be updated.

Thank you for your kind suggestions. Beside the originally cited reviews on formic acid decomposition (*Chem. Soc. Rev.*, 50 (2021) 3437, *J. Mater. Chem. A*, 9 (2021) 24241, *Catal. Rev. Sci. Eng.*, 64 (2021) 835, *Int. J. Hydrog. Energ.*, 43 (2018) 7055, *Catal. Sci. Technol.*, 6 (2016) 12, *Energ. Environ. Sci.*, 5 (2012) 8171), **we have added the other three most relevant recent reviews in the revised piece** (*Mater. Today Chem.*, 26 (2022) 101120, *J. Energ. Chem.*, 70 (2022) 292-309, *Ind. Eng. Chem. Res.*, 61 (2022) 6067-6105). (see Page 34, lines 3-9)

REVIEWERS' COMMENTS

Reviewer #1 (Remarks to the Author):

Following modifications made by the authors, based on the questions raised, I consider the manuscript ready for publication in its current state.

Reviewer #2 (Remarks to the Author):

This reviewer highly values the authors' efforts in addressing the comments and questions raised in my first review. They have effectively resolved most of my initial concerns; however, one of my original concerns still has not been sufficiently addressed.

Use of high operating temperature and noble metals: The vapor phase formic acid decomposition reaction at high operating temperatures (i.e., higher than 100 oC) over the noble metal catalysts would be a great reaction to use for testing the activity of new catalysts and comparison purposes. It could serve as the model reaction like the thermochemical CO oxidation reaction. However, there is no significant importance of such a reaction for the renewable energy field. In this context, the main scientific contribution of the manuscript is its proposed strategy of defect-driven nanostructuring of bimetallic low-nuclearity ensembles (instead of solving the major catalyst issues of the current renewable energy field). Even though their proposed catalyst synthesis process is very interesting, this reviewer cannot clearly understand how this proposed strategy could be used in different catalytic reaction engineering applications and make significant impacts. For this reason, this reviewer believes that this manuscript is not suitable for Nature Communications.

Reviewer #3 (Remarks to the Author):

The authors have addressed most of the issues.

The reviewer is not fully convinced by the presence of “Quantum size effects” (in one case and its absence in another for similar-sized particles), but the argument by the authors cannot be excluded. The revision is accepted.

While stating "significantly outperforming the-state-of-the-art precious metal-based catalysts", the authors may want to mention "under similar reaction conditions comprising the dilute gas-phase reactions" as some of the liquid phase catalysts are performing much better.

Nature Communications – Manuscript number NCOMMS-23-21168A

Response to Reviewers

Comments in blue - Replies in black - Actions in **bold**, citation – in *italic*

Reviewer #1 (Remarks to the Author):

Following modifications made by the authors, based on the questions raised, I consider the manuscript ready for publication in its current state.

We warmly thanked the Reviewer 1 for recommending the publication of this piece.

Reviewer #2 (Remarks to the Author):

This reviewer highly values the authors' efforts in addressing the comments and questions raised in my first review. They have effectively resolved most of my initial concerns; however, one of my original concerns still has not been sufficiently addressed.

The Reviewer 2 is warmly thanked for recognizing our efforts in the last revision and for the further comments.

Use of high operating temperature and noble metals: The vapor phase formic acid decomposition reaction at high operating temperatures (i.e., higher than 100 oC) over the noble metal catalysts would be a great reaction to use for testing the activity of new catalysts and comparison purposes. It could serve as the model reaction like the thermochemical CO oxidation reaction. However, there is no significant importance of such a reaction for the renewable energy field. In this context, the main scientific contribution of the manuscript is its proposed strategy of defect-driven nanostructuring of bimetallic low-nuclearity ensembles (instead of solving the major catalyst issues of the current renewable energy field). Even though their proposed catalyst synthesis process is very interesting, this reviewer cannot clearly understand how this proposed strategy could be used in different catalytic reaction engineering applications and make significant impacts. For this reason, this reviewer believes that this manuscript is not suitable for Nature Communications.

We thanked the Reviewer 2 for praising the merit of the developed strategy of designing the bimetallic low-nuclearity ensembles. As outlined in the Introduction (please kindly refer to **Paragraph 3**), low-nuclearity catalysts are the frontiers in nanomaterial science and have showed great potentials in diverse heterogeneous catalysis. Nonetheless, the currently developed synthesis methods are still quite limited, particularly for the bimetallic ensembles.

“The current synthetic protocols are mainly relying on i) gas-phase/atomic layer depositions that require advanced synthesis facilities and complicated operation procedures⁵⁰, or ii) wet chemistry methods wherein expensive metal carbonyl clusters are often used as the precursors⁴⁵, plus the difficulties in fabricating heteroatom ensembles with structural uniformity. To construct heteroatoms or multi-atom clusters, spatial confinement-pyrolysis strategy has been developed using porous materials like metal-organic frameworks or covalent-organic frameworks to prevent the sintering of different metal precursors, but is mainly limited to a few components such as Fe, Co and Ni⁴⁹.”

The demonstrated defect-driven nanostructuring strategy in this manuscript showed the great potential to prepare a series of bimetallic low-nuclearity Pt-Mo ensembles, and the greatly enhanced catalytic performance in gas-phase formic acid decomposition was achieved owing to the unprecedented Pt₃-Mo₁-N₄ synergy. Altogether, this successful example is expected to stimulate more broad explorations of multiple hetero-atom ensembles for other task-specific catalytic reactions and beyond.

Reviewer #3 (Remarks to the Author):

The authors have addressed most of the issues.

The reviewer is not fully convinced by the presence of “Quantum size effects” (in one case and its absence in another for similar-sized particles), but the argument by the authors cannot be excluded. The revision is accepted.

We sincerely thanked the Reviewer 2 for accepting our revision and for providing additional valuable comments.

While stating "significantly outperforming the-state-of-the-art precious metal-based catalysts", the authors may want to mention "under similar reaction conditions comprising the dilute gas-phase reactions" as some of the liquid phase catalysts are performing much better.

Thank you for your good suggestion. We have revised the sentence accordingly:

*“Notably, the reaction rates of Pt-Mo/NC_{0.07} at 373 and 388 K reached 0.31 and 0.62 mol_{HCOOH} mol_{Pt}⁻¹ s⁻¹, respectively, significantly outperforming the-state-of-the-art precious metal-based catalysts by ca. one order of magnitude **under similar reaction conditions comprising the dilute gas-phase reactions.**”*